# Hippocampal inputs engage CCK+ interneurons to mediate endocannabinoid-modulated feed-forward inhibition in the prefrontal cortex

Xingchen Liu[1], Jordane Dimidschstein[2], Gordon Fishell[2,3], Adam G Carter[1]*

[1]Center for Neural Science, New York University, New York, United States; [2]Stanley Center for Psychiatric Research, Broad Institute of Massachusetts Institute of Technology and Harvard, Boston, United States; [3]Department of Neurobiology, Harvard Medical School, Boston, United States

**Abstract** Connections from the ventral hippocampus (vHPC) to the prefrontal cortex (PFC) regulate cognition, emotion, and memory. These functions are also tightly controlled by inhibitory networks in the PFC, whose disruption is thought to contribute to mental health disorders. However, relatively little is known about how the vHPC engages different populations of interneurons in the PFC. Here we use slice physiology and optogenetics to study vHPC-evoked feed-forward inhibition in the mouse PFC. We first show that cholecystokinin (CCK+), parvalbumin (PV+), and somatostatin (SOM+) expressing interneurons are prominent in layer 5 (L5) of infralimbic PFC. We then show that vHPC inputs primarily activate CCK+ and PV+ interneurons, with weaker connections onto SOM+ interneurons. CCK+ interneurons make stronger synapses onto pyramidal tract (PT) cells over nearby intratelencephalic (IT) cells. However, CCK+ inputs undergo depolarization-induced suppression of inhibition (DSI) and CB1 receptor modulation only at IT cells. Moreover, vHPC-evoked feed-forward inhibition undergoes DSI only at IT cells, confirming a central role for CCK+ interneurons. Together, our findings show how vHPC directly engages multiple populations of inhibitory cells in deep layers of the infralimbic PFC, highlighting unexpected roles for both CCK+ interneurons and endocannabinoid modulation in hippocampal-prefrontal communication.

*For correspondence:
agc5@nyu.edu

Competing interests: The authors declare that no competing interests exist.

## Introduction

The prefrontal cortex (PFC) controls cognitive and emotional behaviors (*Euston et al., 2012*; *Miller and Cohen, 2001*) and is disrupted in many neuropsychiatric disorders (*Godsil et al., 2013*; *Sigurdsson and Duvarci, 2015*). PFC activity is driven and maintained by long-range glutamatergic inputs from a variety of other brain regions (*Hoover and Vertes, 2007*; *Miller and Cohen, 2001*). Strong, unidirectional connections from the ventral hippocampus (vHPC) contribute to both working memory and threat conditioning in rodents (*Jones and Wilson, 2005*; *Sotres-Bayon et al., 2012*; *Spellman et al., 2015*). Dysfunction of vHPC to PFC connectivity is also implicated in schizophrenia, anxiety disorders, chronic stress disorders, and depression (*Godsil et al., 2013*; *Sigurdsson and Duvarci, 2015*). To understand these roles, it is necessary to establish how vHPC inputs engage local excitatory and inhibitory networks within the PFC.

The vHPC primarily projects to the ventral medial PFC in rodents, with axons most prominent in layer 5 (L5) of infralimbic (IL) PFC (*Phillips et al., 2019*). These excitatory inputs contact multiple populations of pyramidal cells and are much stronger at intratelencephalic (IT) cells than nearby pyramidal tract (PT) cells (*Liu and Carter, 2018*). vHPC inputs can drive the robust firing of IT cells

(*Liu and Carter, 2018*), which may be important for maintaining activity during behavioral tasks (*Padilla-Coreano et al., 2016*; *Spellman et al., 2015*). However, they also evoke prominent feed-forward inhibition at pyramidal cells (*Marek et al., 2018*), and excitation and inhibition evolve with different dynamics (*Liu and Carter, 2018*). Here we focus on the mechanisms responsible for this inhibition by establishing which interneurons are engaged by vHPC inputs to the PFC.

As in other cortices, PFC activity is regulated by a variety of GABAergic interneurons, which have distinct functions (*Abbas et al., 2018*; *Courtin et al., 2014*; *Tremblay et al., 2016*). Parvalbumin-expressing (PV+) interneurons mediate feed-forward inhibition via strong synapses at the soma of pyramidal cells (*Atallah et al., 2012*; *Cruikshank et al., 2007*; *Gabernet et al., 2005*). By contrast, somatostatin-expressing (SOM+) interneurons mediate feed-back inhibition via facilitating synapses onto the dendrites (*Gentet et al., 2012*; *Silberberg and Markram, 2007*). However, during trains of repetitive activity, SOM+ interneurons can also participate in feed-forward inhibition (*McGarry and Carter, 2016*; *Tan et al., 2008*). Interestingly, recent in vivo studies suggest both PV+ and SOM+ interneurons may contribute to vHPC-evoked inhibition in the PFC (*Abbas et al., 2018*; *Marek et al., 2018*).

While PV+ and SOM+ interneurons are prominent in the PFC, there is also an unusually high density of cholecystokinin-expressing (CCK+) interneurons (*Whissell et al., 2015*). In the hippocampus, these inhibitory cells help regulate the motivation and emotional state of animals (*Armstrong and Soltesz, 2012*; *Freund, 2003*; *Freund and Katona, 2007*). They also highly express cannabinoid type 1 (CB1) receptors on their axon terminals (*Katona et al., 1999*), and can be strongly modulated by endocannabinoids (*Wilson et al., 2001*; *Wilson and Nicoll, 2001*). For example, brief depolarization of postsynaptic pyramidal cells releases endocannabinoids that bind to CB1 receptors on CCK+ axon terminals and inhibit presynaptic GABA release, a process known as depolarization-induced suppression of inhibition (DSI) (*Wilson and Nicoll, 2001*).

While CCK+ interneurons are prominent in the PFC and may play a role in different forms of inhibition, they remain relatively understudied. A major technical reason is the low-level expression of CCK in pyramidal cells (*Taniguchi et al., 2011*), which makes CCK+ interneurons challenging to specifically target. For example, expressing Cre-dependent reporters in CCK-Cre transgenics also label pyramidal cells in the cortex (*Taniguchi et al., 2011*). Fortunately, this challenge can be overcome with intersectional viruses using the Dlx enhancer, which restricts expression to interneurons (*Dimidschstein et al., 2016*). This approach allows identification of CCK+ interneurons, enabling targeted recordings and optogenetic access to study their connectivity and modulation in the PFC.

Here we examine vHPC-evoked inhibition at L5 pyramidal cells in IL PFC using slice physiology, optogenetics, and intersectional viral tools. We find vHPC inputs activate PV+, SOM+, and CCK+ interneurons, with different dynamics during repetitive activity. Inputs to PV+ and CCK+ interneurons are strong but depressing, while those onto SOM+ interneurons are weak but facilitating. CCK+ interneurons contact L5 pyramidal cells, with stronger connections onto PT cells than neighboring IT cells. However, endocannabinoid modulation via DSI and direct activation of CB1 receptors only occurs at synapses onto IT cells. Endocannabinoid modulation of vHPC-evoked feed-forward inhibition also occurs only at IT cells, highlighting a central role for CCK+ interneurons. Together, our findings show how the vHPC engages interneurons to inhibit the PFC, while revealing a novel property of cell-type-specific endocannabinoid modulation in this circuit.

## Results

### vHPC-evoked inhibition and interneurons in L5 of infralimbic PFC

We studied vHPC-evoked inhibition at pyramidal neurons using whole-cell recordings and optogenetics in acute slices of the mouse PFC. To allow for visualization and activation of inputs to the PFC, we injected the ChR2-expressing virus (AAV-ChR2-EYFP) into the ipsilateral vHPC (*Figure 1A*; *Little and Carter, 2012*). In the same animals, we labeled IT cells by co-injecting retrogradely transported, fluorescently-tagged cholera toxin subunit B (CTB) into the contralateral PFC (cPFC) (*Figure 1A*; *Anastasiades et al., 2018*). After 2–3 weeks of expression, we prepared ex vivo slices of the medial PFC, observing vHPC axons and IT cells in L5 of IL PFC (*Figure 1B*; *Liu and Carter, 2018*). We then recorded in voltage-clamp from IT cells and activated vHPC inputs using wide-field illumination (5 pulses at 20 Hz, 2 ms pulse duration). Trains of vHPC inputs evoked excitatory

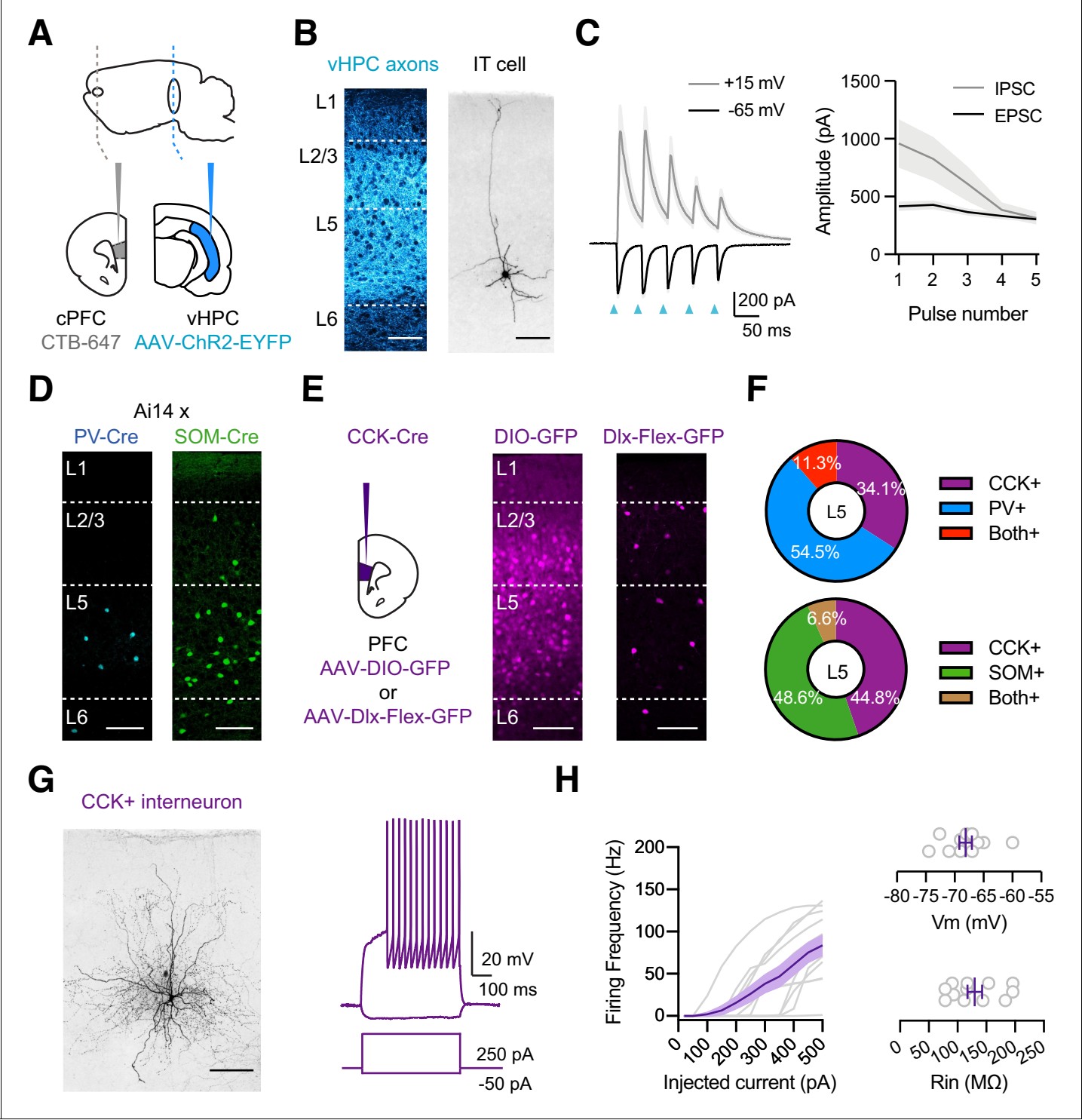

**Figure 1.** vHPC-evoked feed-forward inhibition and CCK+ interneurons. (**A**) Schematic for injections of AAV-ChR2-EYFP into vHPC and CTB-647 into cPFC. (**B**) *Left,* Confocal image of vHPC axons (blue) in IL PFC. Scale bar = 100 μm. *Right,* Confocal image of biocytin-filled L5 IT cell in IL PFC. Scale bar = 100 μm. (**C**) *Left,* Average vHPC-evoked EPSCs at −65 mV (black) and IPSCs at +15 mV (gray). Blue arrows = 5 pulses at 20 Hz. *Right,* Average response amplitudes as a function of pulse number (n = 7 cells, 3 animals). (**D**) Td-tomato labeling of PV+ (blue) and SOM+ (green) interneurons in PV-Cre × Ai14 and SOM-Cre × Ai14 animals, respectively. Scale bar = 100 μm. (**E**) *Left,* Schematic for injections of viruses into PFC of CCK-Cre mice. *Middle,* Injection of AAV-DIO-GFP labels CCK+ interneurons and pyramidal cells (n = 3 animals). *Right,* Injection of AAV-Dlx-Flex-GFP labels CCK+ interneurons (n = 3 animals). Scale bars = 100 μm. (**F**) Quantification of co-labeling of CCK+ interneurons with PV (top) and SOM (bottom) (PV staining, n = 308 cells, 17 slices, 6 animals; SOM staining, n = 105 cells, 8 slices, 3 animals). (**G**) *Left,* Confocal image of a biocytin-filled CCK+ interneuron in L5

*Figure 1 continued on next page*

*Figure 1 continued*

of IL PFC. Scale bar = 100 μm. *Right*, Response to positive and negative current injections. (**H**) *Left*, Firing frequency (F) versus current (I) curve for CCK+ cells. *Right*, Summary of membrane resting potential (Vrest) and input resistance (Rin) of CCK+ interneurons (n = 12 cells, 4 animals).

The online version of this article includes the following figure supplement(s) for figure 1:

**Figure supplement 1.** Anatomy of CCK+ and other interneurons in L5 IL PFC.

postsynaptic currents (EPSCs) at −65 mV and inhibitory postsynaptic currents (IPSCs) at +15 mV (*Figure 1C*; $EPSC_1$ = 386 ± 49 pA, $IPSC_1$ = 961 ± 207 pA, E/I = 0.50 ± 0.08; n = 7 cells, three animals). These findings indicate that vHPC inputs drive robust feed-forward inhibition in deep layers of the IL PFC, motivating us to identify which interneurons are responsible.

In principle, feed-forward inhibition could be mediated by a variety of interneurons, including PV+ and SOM+ interneurons (*Abbas et al., 2018*; *Anastasiades et al., 2018*; *Marek et al., 2018*; *McGarry and Carter, 2016*). To visualize these cells in the PFC, we crossed PV-Cre and SOM-Cre mice with reporter mice (Ai14) that express Cre-dependent tdTomato (*Hippenmeyer et al., 2005*; *Madisen et al., 2010*; *Taniguchi et al., 2011*). We observed labeling of PV+ and SOM+ interneurons in L5 of IL PFC, suggesting they could be contacted by vHPC afferents (*Figure 1D*). However, the PFC also has a high density of cholecystokinin-expressing (CCK+) interneurons (*Whissell et al., 2015*), which mediate inhibition in other cortices and the hippocampus (*Armstrong and Soltesz, 2012*; *Freund and Katona, 2007*), and could also participate in the PFC. To label these cells, we initially injected AAV-DIO-GFP into CCK-Cre mice but observed labeling of both interneurons and pyramidal neurons across multiple layers (*Figure 1E*). To restrict labeling to CCK+ interneurons, we instead used AAV-Dlx-Flex-GFP, expressing Cre-dependent GFP under control of the Dlx enhancer (*Dimidschstein et al., 2016*). Injecting AAV-Dlx-Flex-GFP selectively labeled CCK+ interneurons in the IL PFC, including prominent labeling in L5 (*Figure 1E*). Importantly, we found little co-labeling of CCK+ cells with either PV (*Figure 1F* and *Figure 1—figure supplement 1*; 11.3% overlap, n = 308 cells total, 17 slices, 6 animals) or SOM (6.6% overlap, n = 105 cells total, 8 slices, 3 animals).

To confirm the targeting of CCK+ interneurons, we next used whole-cell recordings followed by post-hoc reconstructions (*Figure 1G*). We found both axons and dendrites in L5 of IL PFC, and intrinsic properties similar to reports in other parts of the brain (*Figure 1H*; Rin = 130 ± 13 MΩ, Vm = −68 ± 1 mV, Sag = 4.1 ± 1.0%, Adaptation = 0.81 ± 0.04, Tau = 9.1 ± 0.6 ms; n = 12 cells, 4 animals) (*Daw et al., 2009*). These results confirm that our viral strategy can identify CCK+ interneurons, which are present in deeper layers of IL, and also show that PV+, SOM+, and CCK+ interneurons are positioned to receive vHPC inputs and may mediate feed-forward inhibition.

## vHPC inputs primarily engage PV+ and CCK+ interneurons

Previous studies suggest that vHPC inputs engage PV+ and SOM+ interneurons in the PFC (*Abbas et al., 2018*; *Marek et al., 2018*). However, these connections have not been examined in L5 of IL, where vHPC connections are the strongest (*Liu and Carter, 2018*). Moreover, little is known about the activation of CCK+ interneurons, which may play distinct functional roles (*Armstrong and Soltesz, 2012*; *Freund and Katona, 2007*). To study connectivity, we injected AAV-ChR2-YFP into the vHPC and recorded EPSCs from identified interneurons. To compare across animals and cell types, we recorded within the same slice from neighboring IT cells, which receive the bulk of vHPC inputs (*Liu and Carter, 2018*). To isolate monosynaptic connections, we included TTX (1 μM), 4-AP (10 μM), and elevated $Ca^{2+}$ (4 mM), which blocks action potentials (APs) but restores presynaptic release (*Little and Carter, 2012*; *Petreanu et al., 2009*). We kept the light intensity and duration constant within pairs, allowing us to account for differences in viral expression across animals and slices (*Liu and Carter, 2018*). We found that vHPC inputs evoked robust EPSCs at CCK+ interneurons, which were similar in amplitude to IT cells (*Figure 2A*; IT = 674 ± 102 pA, CCK+ = 413 ± 89 pA, CCK+/IT ratio = 0.56, p = 0.13; n = 9 pairs, 4 animals). We also found prominent vHPC-evoked EPSCs at PV+ interneurons, with comparable amplitudes at IT cells (*Figure 2B*; IT = 569 ± 91 pA, PV+ = 555 ± 83 pA, PV+/IT ratio = 0.94, p = 0.82; n = 9 pairs, 4 animals). By contrast, vHPC inputs evoked much smaller EPSCs at SOM+ interneurons compared to IT cells (*Figure 2C*; IT = 421 ± 69 pA, SOM+ = 95 ± 33 pA, SOM+/IT ratio = 0.17, p = 0.004; n = 9 pairs, 4 animals). These findings

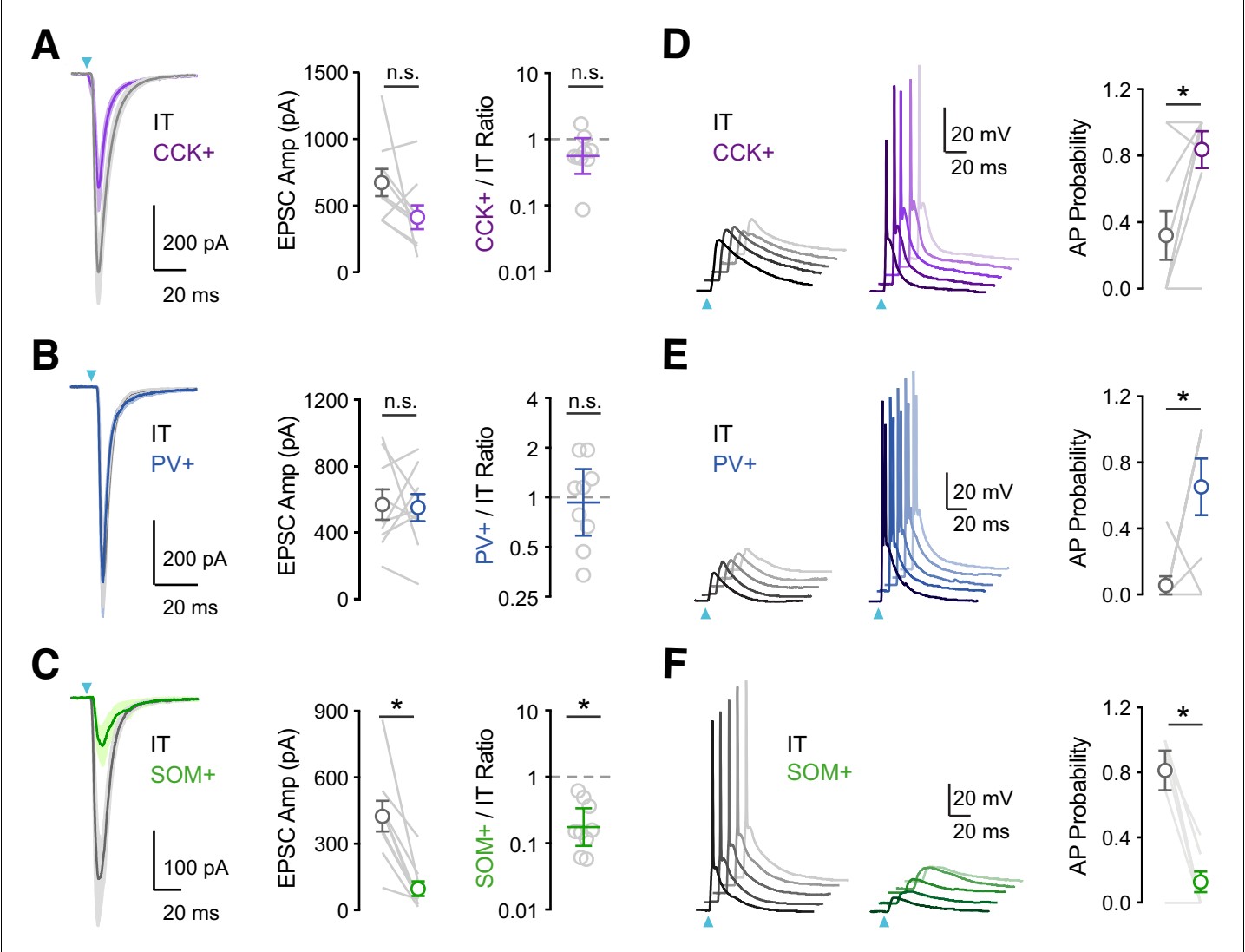

**Figure 2.** vHPC inputs differentially engage PV+, SOM+, and CCK+ interneurons. (A) *Left,* Average vHPC-evoked EPSCs at pairs of L5 IT (gray) and CCK+ (purple) cells in IL PFC. Blue arrow = light pulse. *Middle,* Summary of EPSC amplitudes. *Right,* Summary of CCK+/IT EPSC amplitude ratios (n = 9 pairs, 4 animals). (B – C) Similar to (A) for pairs of IT and PV+ (blue) cells (n = 9 pairs, 4 animals) or pairs of IT and SOM+ (green) cells (n = 9 pairs, 4 animals). (D) *Left,* vHPC-evoked EPSPs and APs recorded in current-clamp from resting membrane potential at pairs of L5 IT (gray) and CCK+ (purple) cells in IL PFC, with 5 traces offset for each cell. *Right,* Summary of AP probability at pairs of IT and CCK+ cells. Blue arrow = 3.5 mW light pulse (n = 9 pairs, 4 animals). (E – F) Similar to (D) for pairs of IT and PV+ cells (3.5 mW light pulses, n = 8 pairs, 4 animals) or pairs of IT and SOM+ cells (4.8 mW light pulses, n = 7 pairs, 3 animals). *p<0.05.

show that all three cell types receive direct vHPC inputs, with greater responses at CCK+ and PV+ interneurons.

Having established the targeting of vHPC inputs, we next assessed their ability to drive action potentials (APs) at the three classes of interneurons. We used similar viral and labeling approaches but in this case conducted current-clamp recordings in the absence of TTX and 4-AP and at physiological $Ca^{2+}$ concentration (2 mM). We also kept light intensity constant within each set of experiments, at a power that evoked action potentials in at least one of the recorded pair of cells. We found that single vHPC inputs evoked APs in pairs of CCK+ and IT cells but the interneurons showed a higher probability of firing (*Figure 2D*; AP probability: IT = 0.32 ± 0.15, CCK+ = 0.84 ± 0.11, p = 0.03; n = 9 pairs, 4 animals). Similarly, we observed that vHPC inputs also preferentially activate PV+ over IT cells (*Figure 2E*; AP probability: IT = 0.06 ± 0.06, PV+ = 0.65 ± 0.17, p = 0.04; n = 8 pairs, 4 animals). By contrast, SOM+ interneurons remained unresponsive to vHPC inputs, even when using

higher light intensities that were able to activate IT cells (*Figure 2F*; AP probability: IT = 0.81 ± 0.12, SOM+ = 0.13 ± 0.06, p = 0.01; n = 7 pairs, 3 animals). These results established a hierarchy for activation, suggesting CCK+ and PV+ interneurons are engaged by vHPC inputs and mediate feed-forward inhibition.

## Short-term dynamics differ between populations of interneurons

Repetitive activity at vHPC to PFC connections is functionally important and depends on stimulus frequency (*Liu and Carter, 2018*; *Siapas et al., 2005*). At the synaptic level, repetitive activity engages short-term plasticity to change the strength of individual connections (*Zucker and Regehr, 2002*). We next examined the response to repetitive vHPC inputs by stimulating with brief trains (5 pulses at 20 Hz) in the absence of TTX and 4-AP (*Liu and Carter, 2018*; *McGarry and Carter, 2016*). We observed that EPSCs at CCK+ and PV+ interneurons strongly depress over the course of stimulus trains (*Figure 3A & B*; $EPSC_2 / EPSC_1$: CCK+ = 0.82 ± 0.06; n = 6 cells, 3 animals; PV+ = 0.88 ± 0.06; n = 7 cells, 3 animals). By contrast, the EPSCs at SOM+ interneurons initially facilitated during trains (*Figure 3A & B*; $EPSC_2 / EPSC_1$ = 1.78 ± 0.23; n = 8 cells, 3 animals). These results indicate that vHPC engages all three cell types, with connections at PV+ and CCK+ interneurons strong but depressing, and those at SOM+ interneurons weak but facilitating.

The short-term dynamics of vHPC inputs suggested differential engagement of PV+, SOM+, and CCK+ interneurons during repetitive activity. We observed that stimulus trains of vHPC inputs generated decreasing AP probabilities at CCK+ and PV+ interneurons (*Figure 3C & D*; stimulus 1 and 2 AP probabilities: CCK+ = 0.97 ± 0.02 and 0.75 ± 0.06; n = 12 cells, 5 animals; PV+ = 0.96 ± 0.02 and

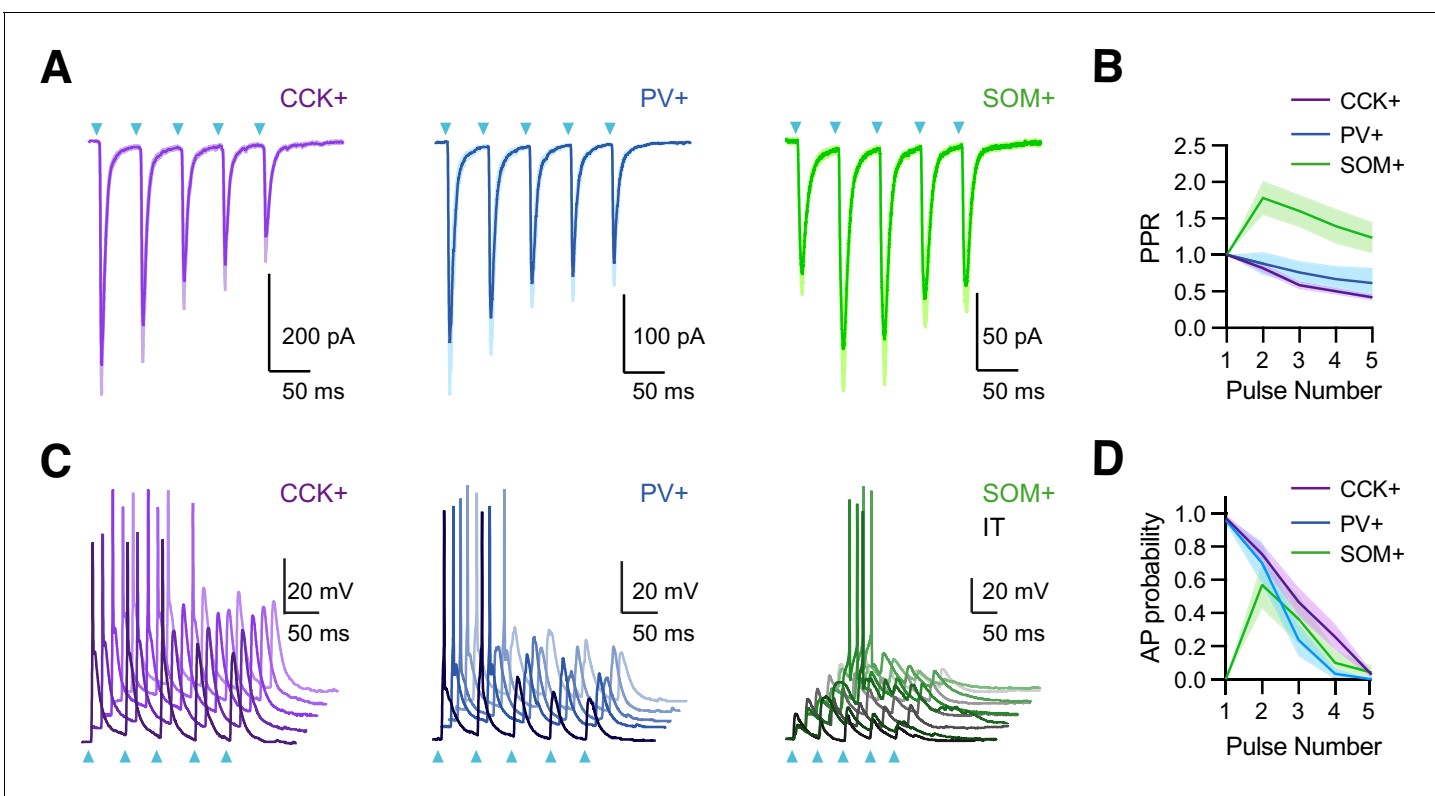

**Figure 3.** vHPC inputs drive interneurons with distinct temporal dynamics. (**A**) *Left*, Average vHPC-evoked EPSCs at CCK+ interneurons, recorded in voltage-clamp at −65 mV (5 pulses at 20 Hz). Blue arrows = light pulses (n = 6 cells, 3 animals). *Middle*, For PV+ interneurons (n = 7 cells, 3 animals). *Right*, For SOM+ interneurons (n = 8 cells, 3 animals). (**B**) Average paired-pulse ratio (PPR) of vHPC-evoked EPSCs at CCK+, PV+, and SOM+ interneurons. (**C**) *Left*, vHPC-evoked EPSPs and APs, recorded in current-clamp from resting membrane potential at an example CCK+ interneuron in L5 of IL PFC, with 5 traces offset for the cell. Blue arrows = light pulses (n = 12 cells, 5 animals). *Middle*, For PV+ interneurons (n = 6 cells, 3 animals). *Right*, For SOM+ interneurons. Note that each SOM+ cell was studied with a nearby IT cell under the same recording and stimulation conditions, where the absence of AP firing at IT cells indicated subthreshold of network recurrent activation (n = 6 cells, 3 animals). (**D**) Summary of average AP probability as a function of pulse number for CCK+, PV+, and SOM+ interneurons.

0.70 ± 0.13; n = 6 cells, 3 animals;). By contrast, although SOM+ interneurons did not fire with single pulses of vHPC inputs, they were activated during trains. Importantly, these responses were not due to recurrent network activity, as stimulation intensity was chosen here to ensure that IT cells remained quiescent (*Figure 3C & D*; stimulus 1 and 2 AP probabilities: IT = 0 and 0, SOM+ = 0 and 0.57 ± 0.14; n = 6 pairs, 4 animals). These findings suggest that vHPC differentially engages three interneuron types with distinct dynamics during repetitive activity, with strong and depressing PV+ and CCK+ activity contributing early, and weaker but facilitating SOM+ activity contributing later.

## CCK+ interneurons make connections onto L5 pyramidal cells

Our results indicate that vHPC inputs strongly engage CCK+ interneurons in the PFC, suggesting a role in feed-forward inhibition. However, the connections made by CCK+ interneurons onto different pyramidal cell subtypes are not well established in the PFC. To study CCK+ output, we developed a new virus (AAV-Dlx-Flex-ChR2-mCherry) to express ChR2 in a Cre-dependent manner under the Dlx enhancer (see Materials and methods). We injected this virus into the PFC of CCK-Cre mice to selectively express ChR2 in CCK+ interneurons (*Figure 4A*). Whole-cell current-clamp recordings showed that labeled cells could be rapidly and reliably activated by brief pulses of blue light, with a single AP elicited for each pulse during the train (*Figure 4A*; stimulus 1 to 5, AP number: 1.0 ± 0.0, n = 6 cells, 3 animals). To study inhibitory connections, we then recorded CCK+-evoked currents from unlabeled L5 pyramidal cells in IL PFC (*Figure 4B*). To detect both EPSCs or IPSCs, we used a low chloride internal and held at −50 mV, such that inward currents were EPSCs and outward currents were IPSCs (*Glickfeld and Scanziani, 2006*). We observed robust CCK+-evoked IPSCs, which were unaffected by blockers of AMPAR (10 μM NBQX) and NMDAR (10 μM CPP) but abolished by blockers of GABA$_A$R (10 μM gabazine) (*Figure 4B*; ACSF = 97 ± 21 pA, NBQX+CPP = 97 ± 21 pA, gabazine = 0.4 ± 0.2 pA; ACSF versus NBQX + CPP, p = 0.69; NBQX + CPP versus gabazine, p = 0.01; n = 7 cells, 3 animals). These findings indicate that CCK+ interneurons make inhibitory connections onto neighboring L5 pyramidal cells and that our viral strategy avoids contamination from excitatory contacts due to inadvertent activation of CCK+ pyramidal cells.

Previous studies indicate that inhibitory inputs from PV+ and SOM+ interneurons are strongly biased onto PT cells over nearby IT cells (*Anastasiades et al., 2018*). To test if similar biases occur for CCK+ interneurons, we labeled PT and IT cells by injecting retrograde tracers into periaqueductal gray (PAG) and cPFC, respectively (*Figure 4C*). Recording from pairs of pyramidal cells, we found that CCK-evoked IPSCs were larger onto PT cells than IT cells (*Figure 4D*; IPSC IT = 113 ± 59 pA, PT = 245 ± 79 pA, p = 0.01; n = 7 pairs, 4 animals). These findings indicate that CCK+ interneurons make cell-type specific connections, preferentially targeting PT cells in L5 of the IL PFC.

Related studies on the hippocampus show that pyramidal cell innervation by CCK+ interneurons can vary along the somato-dendritic axis (*Lee et al., 2010*). We next used subcellular channelrhodopsin assisted circuit mapping (sCRACM) to study the subcellular targeting of CCK+ interneurons onto defined projection neurons in the PFC (*Petreanu et al., 2009*). We expressed ChR2 in CCK+ interneurons and recorded from retrogradely labeled IT and PT cells in the presence of 1 μM TTX and 10 μM 4-AP to enable terminal activation and isolate monosynaptic inputs (*Little and Carter, 2012*). We activated CCK+ inputs across the entire somato-dendritic axis using a grid (10 × 10) of pseudorandomly delivered (1 Hz) spots of light (75 μm diameter) (*Figure 4—figure supplement 1A & C*). We found CCK+ inputs are highly restricted to the somatic compartment of both IT and PT cells (*Figure 4—figure supplement 1B & D*). Therefore, in contrast to the strong dendritic innervation by SOM+, 5HT3aR+, and NDNF+ cells (*Anastasiades et al., 2020*; *Marlin and Carter, 2014*), these results indicate that CCK+ inputs are primarily near the soma of IT and PT cells.

## Expression specificity of CB1Rs

Throughout the brain, CCK+ interneurons are distinguished from PV+ cells by enrichment of the cannabinoid type one receptor (CB1R) on presynaptic terminals (*Bodor et al., 2005*; *Dudok et al., 2015*; *Katona et al., 1999*). To examine the expression of CB1R in the IL PFC, we next used immunocytochemistry and detected CB1R+ puncta located at PV+ and CCK+ axons. We found CB1R+ puncta showed higher co-localization with CCK+ axons (based on GFP+ expression) than with PV+ axons (*Figure 4E–G*; puncta density in 10³/μm²: PV+CB1R+ = 3.79 ± 0.64, GFP+CB1R+ = 11.27 ±

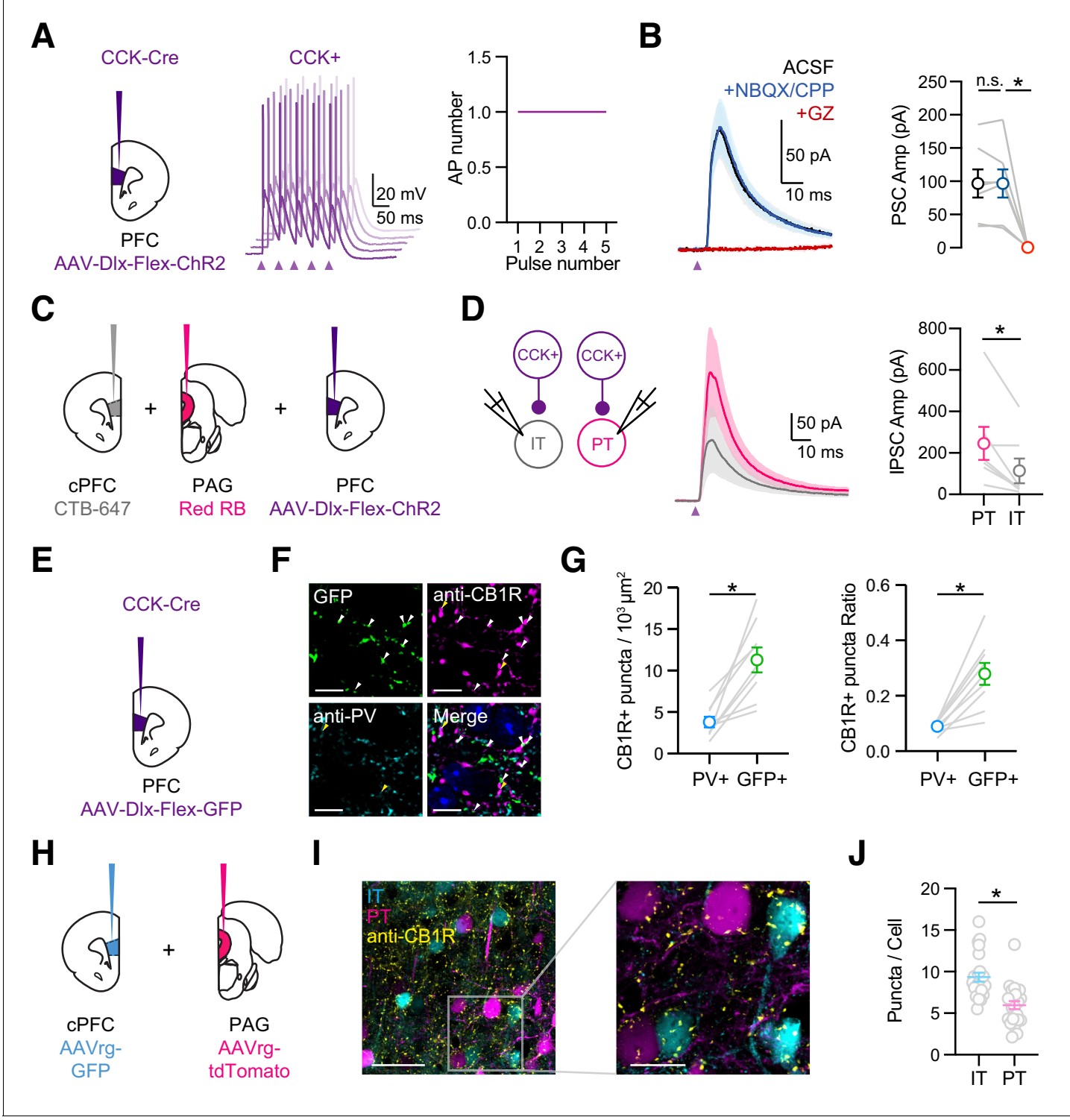

**Figure 4.** CCK+ interneurons make connections onto pyramidal cells. (A) *Left*, Injections of AAV-Dlx-Flex-ChR2 into the PFC of CCK-Cre animals. *Middle*, Example AP traces from a ChR2-expressing CCK+ interneuron, stimulated with 2 ms light pulses at 20 Hz, showing 5 offset trials. Purple arrow = light pulse. *Right*, Average AP numbers across stimulation pulses (n = 6 cells, 3 animals). (B) *Left,* Average CCK+-evoked IPSCs at L5 pyramidal cells in IL PFC. When recording at −50 mV with a low Cl- internal solution, only outward IPSCs were observed (black). IPSCs were unchanged after wash-in of NBQX + CPP (blue) but abolished by wash-in of gabazine (red). *Right*, Summary of IPSC amplitudes in the different conditions. Purple arrow = light pulse. (C) Schematic of triple injections, with CTB-647 into cPFC, red retrobeads into PAG, and AAV-Dlx-Flex-ChR2 into PFC. (D) *Left*, Recording schematic of CCK+ inputs onto IT and PT cells. *Middle*, CCK+-evoked IPSCs at PT and IT cells. *Right*, Summary of IPSC amplitudes at PT and IT cells (7

*Figure 4 continued on next page*

*Figure 4 continued*

cells, 4 animals). (E) Injection schematic. (F) Confocal images of GFP (green), anti-CB1R staining (purple), anti-PV staining (cyan), and merge. Blue labeling in merged image = DAPI. Arrow heads: white = GFP+CB1R+ co-labeling, yellow = PV+CB1R+ co-labeling. (G) *Left,* Quantification of PV+CB1R+ and GFP+CB1R+ quanta per $10^3$ µm². *Right,* Quantification of the ratios of CB1R+ puncta among PV+ and GFP+ puncta. Each line represents counts from one slice (n = 9 slices, 3 animals). (H) Injection schematic of AAVrg-GFP into PFC and AAVrg-tdTomato into PAG. (I) *Left,* Confocal image of IT cells (cyan), PT cells (magenta), and CB1 receptors (yellow). Scale bar = 50 µm. *Right,* Magnification of region on left. Scale bar = 20 µm. (J) Quantification of CB1R puncta in IT and PT cells, each dot represents the average puncta number per cell in a slice (n = 247 IT cells, 207 PT cells, 4 animals). *p<0.05.

The online version of this article includes the following figure supplement(s) for figure 4:

**Figure supplement 1.** Subcellular targeting of CCK+ interneurons to IT and PT cells in L5 IL PFC.

1.5, p=0.0003). These results suggest that CCK+ interneurons are enriched with presynaptic CB1Rs and are likely to undergo endocannabinoid modulation.

Having already established that CCK+ outputs preferentially target PT cells, we next examined if CB1R+ expression is cell-type specific. We labeled IT and PT cells by injecting retrogradely transported AAVrg-TdTomato into the cPFC and AAVrg-GFP into the PAG, respectively. We then used immunocytochemistry to examine CB1R+ puncta surrounding the cell bodies of neighboring IT and PT cells (*Figure 4H*). While we observed CB1R+ puncta around the cell bodies of both cell types (*Figure 4I*), their density was much greater at IT cells (*Figure 4J*; IT = 9.3 ± 0.5 puncta/cell, PT = 5.7 ± 0.5 puncta/cell, p<0.0001; n = 4 animals, 247 IT cells, 207 PT cells). These results indicate that presynaptic CB1Rs are more prominent at perisomatic connections onto IT cells, suggesting that endocannabinoid modulation may be more extensive at those projection neurons.

## CB1R-mediated DSI depends on the postsynaptic cell-type

In the hippocampus and the amygdala, the strength of CCK+ inputs to pyramidal cells is strongly modulated by endocannabinoids (*Lee et al., 2010*; *Vogel et al., 2016*; *Wilson and Nicoll, 2001*). Postsynaptic depolarization releases endocannabinoids that act on presynaptic CB1 receptors to prevent GABA release, a process known as depolarization-induced suppression of inhibition (DSI) (*Wilson and Nicoll, 2001*). To study DSI at IT and PT cells, we injected retrogradely transported CTBs in the cPFC and PAG, along with AAV-Dlx-Flex-ChR2 in the PFC of CCK-Cre mice (*Figure 5A*). We evoked DSI with a standard protocol (*Glickfeld and Scanziani, 2006*; *Wilson and Nicoll, 2001*), recording baseline CCK+-evoked IPSCs during a train, followed by a 5 s depolarization to +10 mV and recording CCK+-evoked IPSCs again after an initial 1 s delay, followed by a recovery test after another 30 s delay (*Figure 5B*). For these experiments, it was critical to record IPSCs at −50 mV rather than +10 mV, which allows us to control the timing of endocannabinoid release. We also used a low-Cl⁻ internal solution, with the Cl⁻ conductance reversing at −80 mV, in order to increase the driving force for GABAa-R currents. In voltage-clamp recordings from retrogradely labeled IT cells, we found that CCK+-evoked IPSCs underwent pronounced DSI that recovered to baseline when inputs were stimulated again 30 s later (*Figure 5C & F*; $IPSC_1$: before = 158 ± 34 pA, after = 91 ± 24 pA, recovery = 149 ± 33 pA; DSI ratio = 0.57 ± 0.06, p = 0.004; n = 9 cells, 5 animals), with these recovery kinetics similar to previous reports in other preparations (*Figure 5—figure supplement 1*; n = 10 cells, four animals) (*Glickfeld and Scanziani, 2006*). We confirmed the involvement of endocannabinoid signaling by blocking DSI with the CB1R inverse agonist 10 µM AM-251 (*Figure 5D & F*; $IPSC_1$: before = 189 ± 53 pA, after = 168 ± 44 pA, recovery = 185 ± 55; DSI ratio = 0.92 ± 0.04, p = 0.08; n = 7 cells, 4 animals). We also confirmed this modulation was specific to CCK+ inputs, as PV+ and SOM+-evoked IPSCs at IT cells had minimal DSI (*Figure 5F* and *Figure 5—figure supplement 2*; $IPSC_1$ DSI ratio: PV+ = 0.94 ± 0.02, p = 0.11; n = 7 cells, 3 animals; SOM+ = 0.87 ± 0.05, p = 0.02; n = 8 cells, 3 animals; PV+ versus CCK+, p = 0.0007; SOM+ versus CCK+, p = 0.008). Surprisingly, we found minimal DSI at PT cells, indicating modulation also strongly depends on postsynaptic cell type (*Figure 5E & F*; $IPSC_1$ before = 208 ± 40 pA, after = 202 ± 44 pA, recovery = 205 ± 40 pA; DSI ratio = 0.95 ± 0.04, p = 0.22; n = 7 cells, 3 animals). These findings indicate that although CCK+ interneurons broadly contact L5 pyramidal cells, the connections undergo prominent CB1R-mediated DSI only at IT cells and not nearby PT cells.

To test if equivalent CB1R modulation also occurs under more physiological conditions, we performed related current-clamp recordings. We held IT cells at −50 mV to create a driving force and

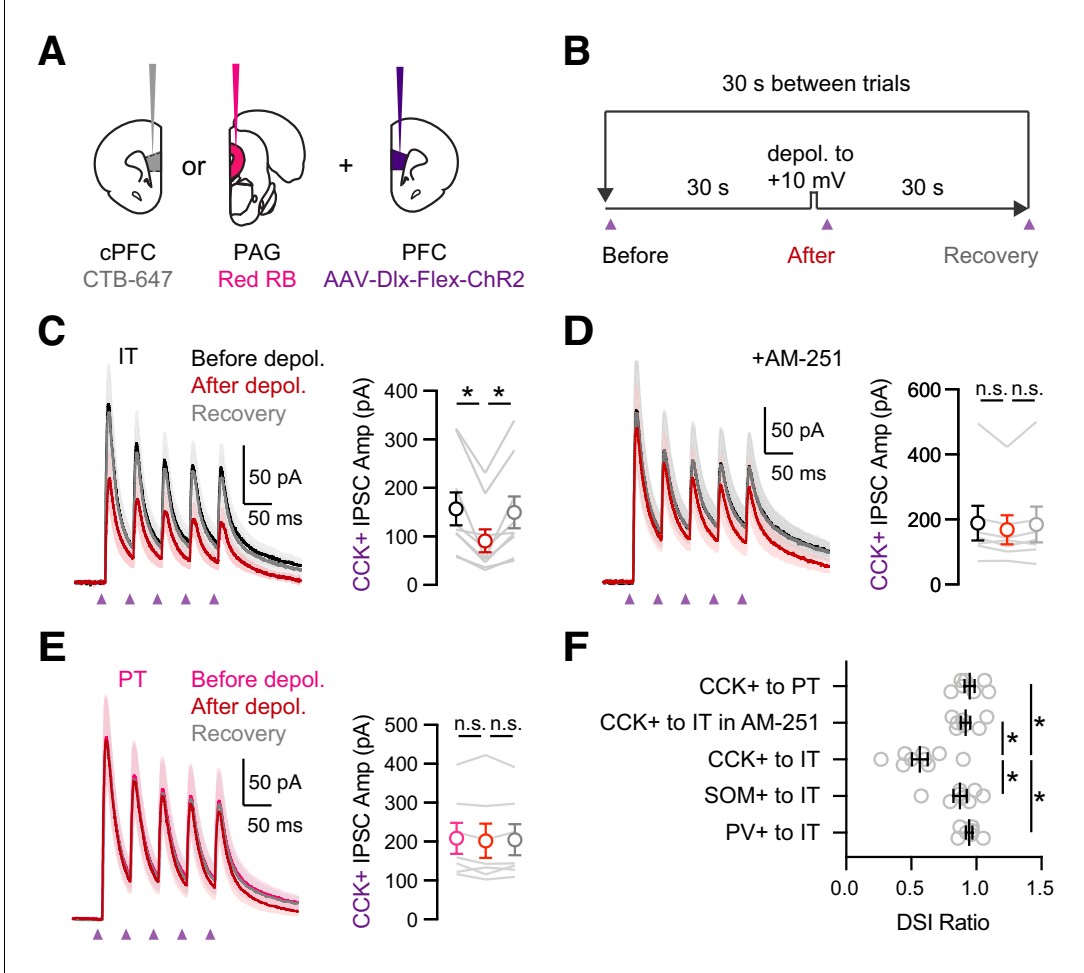

**Figure 5.** Prominent DSI of CCK+ inputs onto IT cells. (A) Injection schematic of CTB-647 into cPFC and red retrobeads (RB) into PAG, along with AAV-Dlx-Flex-ChR2 into PFC of CCK-Cre mice. (B) Experimental procedure for depolarization-induced suppression of inhibition (DSI), with 30 s baseline, followed by 5 s depolarization to +10 mV, and 30 s recovery, repeated every 30 s. (C) *Left*, Average CCK+-evoked IPSCs at IT cells before (black), after (red), and recovery (gray) from the brief depolarization. Purple arrows = 2 ms pulses at 20 Hz. *Right*, Summary of amplitudes of the first IPSC, showing robust DSI (n = 9 cells, 5 animals). (D) Similar to (C) in the presence of 10 μM AM-251, which abolished DSI (n = 7 cells, 4 animals). (E) Similar to (C) for CCK+-evoked IPSCs at PT cells, showing no DSI (n = 7 cells, 3 animals). (F) Summary of DSI ratios (IPSC after depolarization/IPSC before depolarization) across experiments in (C – E). *p<0.05.

The online version of this article includes the following figure supplement(s) for figure 5:

**Figure supplement 1.** Time course of CCK inputs to IT cells during DSI protocol.
**Figure supplement 2.** Minimal DSI at PV+ or SOM+ interneuron connections onto IT cells.
**Figure supplement 3.** Suppression of inhibition induced by IT cell firing.

enable measurement of CCK+-evoked inhibitory postsynaptic potentials (IPSPs) (*Figure 5—figure supplement 3A*). Further depolarization for 5 s evoked AP firing and reduced CCK+-evoked IPSPs at IT cells (*Figure 5—figure supplement 3B & D*; before APs = 11.3 ± 1.5 mV, after APs = 7.7 ± 1.1 mV, recovery = 11.8 ± 1.4 mV; before versus. after APs, p = 0.015; n = 7 cells, 4 animals). However, an additional hyperpolarization also occurred after the APs, which could reduce the driving force for any IPSPs. To account for this possibility, we also recorded CCK+-evoked IPSPs at the matched hyperpolarized membrane potential for each cell and observed intermediate responses (*Figure 5—figure supplement 3B & D*; corrected = 9.0 ± 1.0 mV; corrected versus. after APs, p = 0.015). Finally, we repeated these experiments in the presence of AM-251, which eliminated the difference between these IPSPs (*Figure 5—figure supplement 3C & E*; after APs = 5.1 ± 1.0 mV, corrected = 5.0 ± 0.9 mV, p = 0.9; n = 9 cells, 4 animals). Together, these experiments indicate that equivalent CB1R modulation of CCK+ inputs to IT cells also occurs under more physiological conditions.

## Endocannabinoid modulation depends on postsynaptic cell type

If the cell-type specificity of DSI depends on presynaptic factors, we would expect to observe equivalent differences for pharmacologically evoked endocannabinoid modulation by activating presynaptic CB1Rs. We further examined the mechanism of CB1R modulation using wash-in of the agonist WIN 55,212–1 (WIN, 1 μM) followed by the inverse agonist AM-251(10 μM), which act directly at presynaptic CB1Rs located on CCK+ axon terminals. In voltage-clamp recordings from L5 IT cells, we found that WIN reduced CCK+-evoked IPSCs, which was reversed by AM-251 (*Figure 6A & C*; baseline = 234 ± 53 pA, WIN = 106 ± 26 pA, AM-251 = 206 ± 50 pA; baseline versus WIN, p = 0.008; WIN versus AM-251, p = 0.016; n = 8 cells, 5 animals). Previous studies indicate that CB1Rs might be tonically active and inhibit CCK+ outputs at baseline (*Losonczy et al., 2004*; *Neu et al., 2007*). Recording at IT cells, we found that washing in AM-251 alone slightly increased the CCK+ inputs, confirming weak tonic CB1R activation (*Figure 6—figure supplement 1*; baseline = 186.7 ± 32 pA, AM-251 = 204.1 ± 29 pA, AM-251/baseline = 1.18 ± 0.10, p = 0.01, n = 8 cells, 5 animals). By contrast, recordings at PT cells, we found that neither WIN nor AM-251 had any effect on CCK+-evoked IPSCs, consistent with a lack of DSI (*Figure 6B & C*; baseline = 292 ± 55 pA, WIN = 250 ± 36 pA, AM-251 = 281 ± 37 pA; baseline versus. WIN, p = 0.81; WIN versus AM-251, p = 0.22; n = 7 cells, 4 animals). Overall, the activation of CB1 receptors strongly reduced CCK+-evoked IPSCs only at IT cells but not neighboring PT cells (*Figure 6C*; WIN/baseline: IT = 0.44 ± 0.05, PT = 0.92 ± 0.09, p = 0.001). Interestingly, during trains of CCK+ stimulation, wash-in of WIN also increased the paired-pulse ratio (PPR) at IT cells, suggesting presynaptic modulation of release probability by CB1Rs

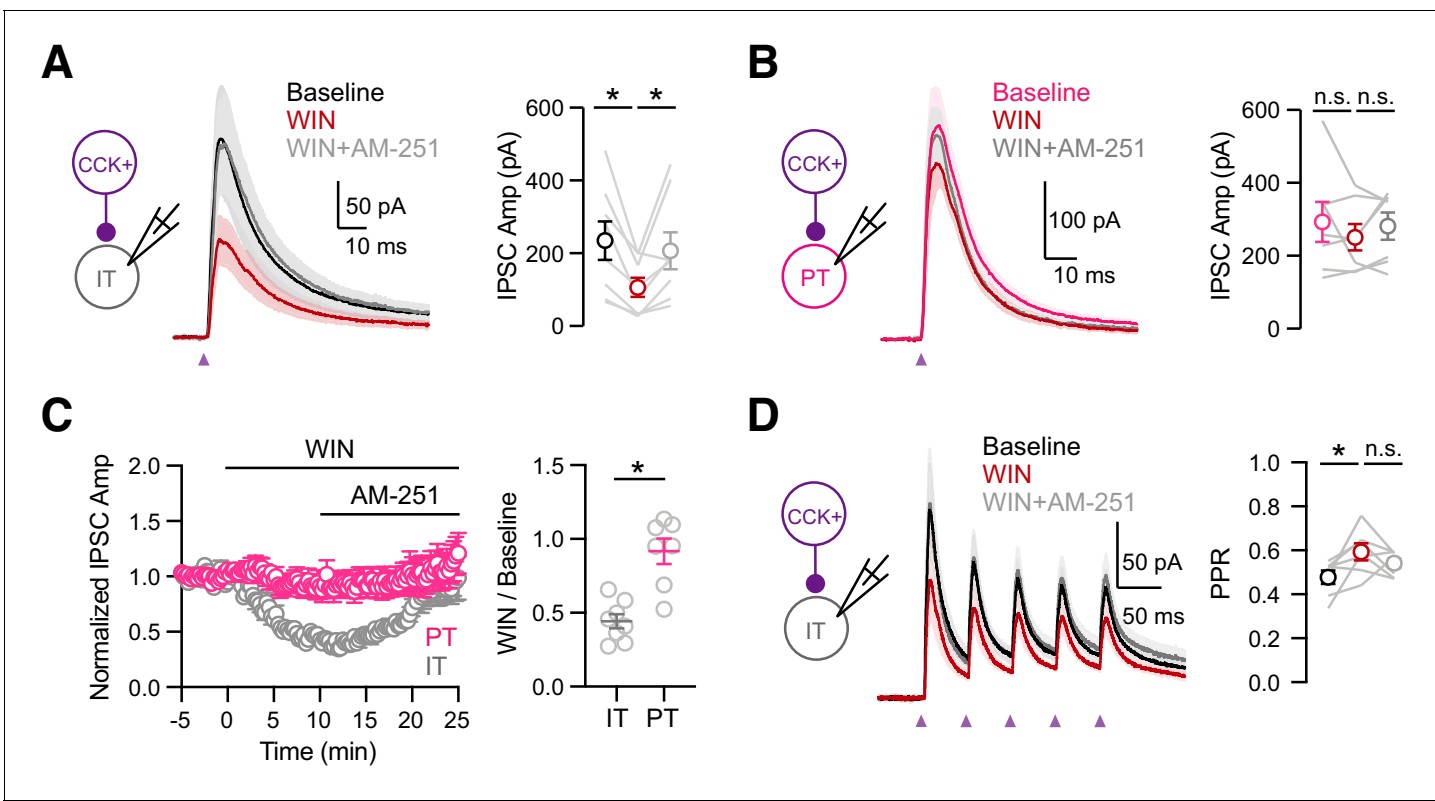

**Figure 6.** Cell-type specific modulation by CB1 receptors. (**A**) *Left,* Schematic of recordings from IT cells in L5 of IL PFC. *Middle,* Average CCK+-evoked IPSCs at IT cells at baseline (black), 10 min after wash-in of 1 μM WIN 55,212–2 (red), and 15 min after additional wash-in of 10 μM AM-251 (gray). Purple arrow = light stimulation. *Right,* Summary of IPSC amplitudes (n = 8 cells, 5 animals). (**B**) Similar to (**A**) for CCK+-evoked IPSCs at PT cells, showing lack of modulation by CB1R (n = 7 cells, 4 animals). (**C**) *Left,* Summary of time course of modulation at IT and PT cells, with IPSC amplitudes normalized to the average response during the first 5 min. *Right,* Summary of normalized IPSC amplitudes after WIN wash-in. (**D**) Similar to (**C**) for trains of CCK+ inputs onto IT cells (5 pulses at 20 Hz), showing small increase in PPR after wash-in of 1 μM WIN 55,212–2 (n = 7 cells, 4 animals). *p<0.05.

The online version of this article includes the following figure supplement(s) for figure 6:

**Figure supplement 1.** The effect of AM-251 on CCK+ inputs to IL L5 IT cells.

(*Figure 6D*; $IPSC_2 / IPSC_1$: baseline = 0.48 ± 0.03, WIN = 0.59 ± 0.03, AM-251 = 0.54 ± 0.02; baseline versus WIN, p = 0.03; WIN versus AM-251, p = 0.22; n = 7 cells, 4 animals). These findings demonstrate that differences in presynaptic endocannabinoid signaling can account for the differential presence of DSI at CCK+ connections onto IT and PT cells. vHPC-evoked feed-forward inhibition is modulated by endocannabinoids.

Together, our results suggest that vHPC inputs strongly engage CCK+ interneurons that in turn robustly inhibit IT cells. Based on the presence of CB1R modulation and DSI at IT cells, we hypothesized that vHPC-evoked inhibition should also undergo target-specific DSI. In voltage-clamp recordings from IT cells, we found that repetitive activation of vHPC inputs evoked EPSCs and IPSCs at −50 mV (*Figure 7A & B*). Depolarization of IT cells (5 s to +10 mV) reduced vHPC-evoked IPSCs but not EPSCs at −50 mV (*Figure 7B & E*; before, after, recovery: $IPSC_1$ = 102 ± 14 pA, 51 ± 10 pA, 88 ± 10 pA; $EPSC_1$ = 248 ± 54 pA, 233 ± 48 pA, 241 ± 52 pA; DSI ratio = 0.47 ± 0.05, p = 0.02; DSE ratio = 0.99 ± 0.06, p = 0.30; n = 7 cells, 4 animals). These findings indicate that there is no DSE at the vHPC to PFC connection, whereas there is prominent DSI onto IT cells. Importantly, application of AM-251 minimized DSI, indicating that it is mediated by endocannabinoids activating CB1Rs (*Figure 7C & E*; $IPSC_1$: before = 136 ± 31 pA, after = 116 ± 29 pA, recovery = 149 ± 35 pA; DSI ratio = 0.84 ± 0.03, p = 0.02; n = 7 cells, 3 animals; DSI in ACSF versus AM-251, p = 0.0006). In contrast, depolarization of PT cells had minimal effect on either vHPC-evoked IPSCs or EPSCs, confirming no DSI or DSE onto this postsynaptic cell type (*Figure 7D & E*; $IPSC_1$ before = 129 ± 32 pA, after = 114 ± 27 pA, recovery = 134 ± 33 pA; DSI ratio = 0.91 ± 0.05, p = 0.10; DSE ratio = 0.89 ± 0.04, p = 0.01; n = 10 cells, 4 animals). These findings show that DSI of vHPC-evoked inhibition only occurs at IT cells, suggesting that CCK+ interneurons are a key node for activity-dependent modulation in communication between vHPC and PFC.

## Discussion

We have explored several new features related to the organization and modulation of connections from vHPC to PFC (*Figure 7F*). First, we found vHPC contacts and strongly activates CB1R-expressing CCK+ interneurons in L5 of IL PFC. Second, we showed that CCK+ interneurons contact nearby pyramidal cells, suggesting they participate in feed-forward inhibition. Third, we found that CCK+ connections undergo CB1R-mediated modulation and DSI, which is selective for IT and not PT cells. Fourth, endocannabinoids also modulate vHPC-evoked inhibition, which also undergoes DSI selectively at IT cells. Together, our results reveal a central role for CCK+ interneurons and endocannabinoid modulation in communication between vHPC and PFC.

The PFC possesses a rich variety of GABAergic interneurons, which are known to have unique roles in goal-directed behaviors (*Abbas et al., 2018*; *Courtin et al., 2014*; *Kepecs and Fishell, 2014*; *Kvitsiani et al., 2013*). Interestingly, the PFC has fewer PV+ interneurons and more CCK+ interneurons compared to other cortices (*Kim et al., 2017*; *Whissell et al., 2015*). Our results indicate that CCK+ interneurons are abundant in L5 of IL PFC, which we previously showed receives the strongest connections from vHPC (*Liu and Carter, 2018*). We found these cells are distinct from PV+ and SOM+ interneurons, with different morphological and physiological properties. We also found that they are enriched in presynaptic CB1Rs, as in other parts of cortex, hippocampus, and amygdala (*Armstrong and Soltesz, 2012*; *Bodor et al., 2005*; *Vogel et al., 2016*). The abundance of CCK+ interneurons in the ventral mPFC is consistent with their role in cognitive and emotional behaviors (*Freund, 2003*; *Freund and Katona, 2007*). In the future, it will be interesting to characterize CCK+ interneurons in other layers and subregions of the PFC.

One of our key results is that vHPC inputs densely contact and strongly activate CCK+ interneurons in L5 of IL PFC. vHPC inputs are strong and depressing onto CCK+ interneurons, in contrast to the facilitating inputs onto IT cells (*Liu and Carter, 2018*). Importantly, vHPC-evoked firing of CCK+ interneurons occurs without activation of IT cells, indicating polysynaptic recurrent activity is not required for activation of CCK+ interneurons, which are therefore likely to mediate feed-forward inhibition. With higher intensity of input stimulation, CCK+ interneurons may also be activated by local inputs, as observed in the hippocampus (*Glickfeld and Scanziani, 2006*), and similar to SOM+ interneurons in the cortex (*Silberberg and Markram, 2007*). Indeed, in the hippocampus, CCK+ interneurons have been shown to participate in both feed-forward and feed-back networks (*Basu et al., 2013*; *Glickfeld and Scanziani, 2006*). Our findings show that CCK+ interneurons have

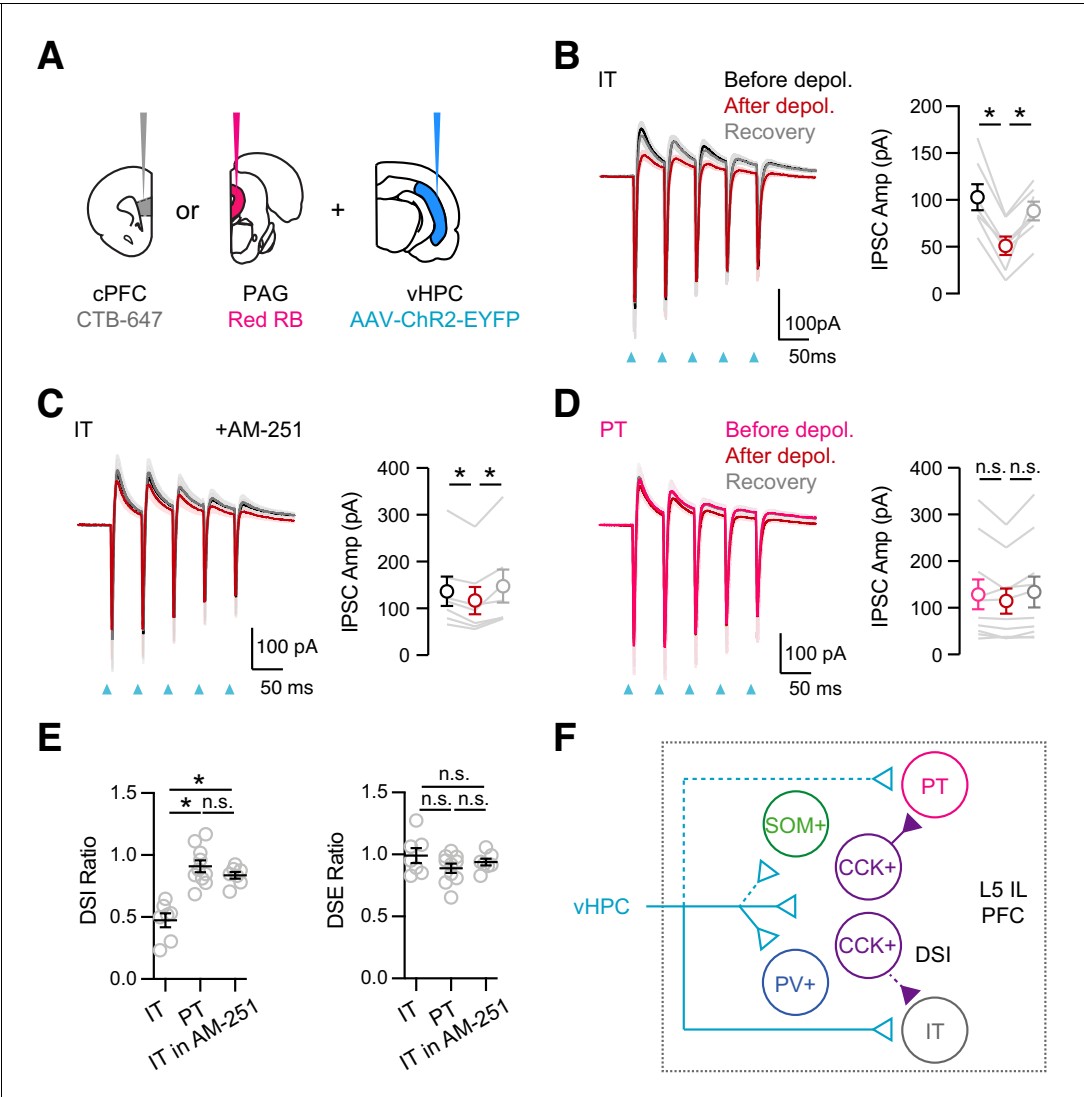

**Figure 7.** vHPC-evoked feed-forward inhibition at IT cells undergoes DSI. (A) Injection schematic, showing CTB-647 in cPFC, red retrobeads (RB) in PAG, along with AAV-ChR2-EYFP in vHPC. (B) *Left*, Average vHPC-evoked EPSCs and IPSCs at IT cells in L5 IL PFC, before (black), immediately after depolarization (red), and after recovery (gray) (same paradigm as *Figure 5*). Blue arrows = light stimulation. *Right*, Summary of amplitudes of first vHPC-evoked IPSCs (n = 7 cells, 4 animals). (C) Similar to (B) in the presence of 10 μM AM-251, which reduces DSI (n = 7 cells, 3 animals). (D) Similar to (B) for PT cells, showing no DSI (n = 10 cells, 4 animals). (E) Summary of DSI and DSE ratios (amplitude ratios of IPSCs or EPSCs after/before the depolarizations) across the different experiments. (F) Summary schematic for vHPC-evoked feed-forward inhibition in IL PFC. vHPC inputs directly contact IT over PT cells to evoke EPSCs. vHPC inputs also engage multiple interneurons to evoke local inhibition. Inhibition mediated by CCK+ interneurons displays robust endocannabinoid-mediated DSI, but only at IT cells, and not neighboring PT cells. *p<0.05.

an important and underappreciated role in hippocampal evoked feed-forward inhibition, and it will be important to assess whether this generalizes to other inputs to the PFC.

While we focused on CCK+ interneurons, we also confirmed that vHPC inputs engage other interneurons in L5 of IL PFC. The activation of PV+ interneurons is strong, consistent with their role in feed-forward inhibition in PFC and elsewhere (*Anastasiades et al., 2018*; *Cruikshank et al., 2007*; *Delevich et al., 2015*; *Gabernet et al., 2005*; *McGarry and Carter, 2016*). Previous results have indicated that vHPC engages PV+ interneurons in superficial layers of the medial PFC (*Marek et al., 2018*). By contrast, we find particularly strong connections in L5, where our previous study shows vHPC inputs are most dense (*Liu and Carter, 2018*). In the future, it will also be interesting to address the relative contribution of PV+ and CCK+ interneurons in feed-forward inhibition, which may be accomplished by novel genetic tools that allow for the labeling of CCK+ and PV+

interneurons in the same animal (*Fenno et al., 2014*). SOM+ interneuron activation is weak, but increases with repetitive activity, similar to BLA inputs to superficial PFC (*McGarry and Carter, 2016*). The engagement of SOM+ interneurons is consistent with a role in oscillations linking the vHPC and PFC and involvement in working memory (*Abbas et al., 2018*). Interestingly, the activation of SOM+ interneurons by long-range inputs also occurs in the granular sensory cortex, where it also builds during stimulus trains, suggesting this is a general property (*Tan et al., 2008*).

Our intersectional viral approach enabled cell-type specific identification and optogenetic activation of cortical CCK+ interneurons. We found that CCK+ interneurons make robust inhibitory connections onto neighboring pyramidal cells in L5 of IL PFC. Importantly, these connections are much stronger onto PT cells, similar to our previous findings for PV+, SOM+, and NDNF+ interneurons (*Anastasiades et al., 2020*; *Anastasiades et al., 2018*). Biased inhibition is thus a general property of inhibitory connections in the PFC, although strength also depends on intrinsic properties (*Anastasiades et al., 2018*). In the hippocampus, CCK+ interneurons also make unique connections onto different pyramidal cell populations defined by sublayer (*Valero et al., 2015*). While hippocampal CCK+ innervations vary along the somato-dendritic axis of pyramidal cells (*Bodor et al., 2005*; *Lee et al., 2010*), our sCRACM results suggest CCK+ interneurons target the soma of both IT and PT cells in L5 of IL PFC. This subcellular targeting could explain why we observed little asynchronous CCK+ release, which is stronger at dendritic synapses compared to somatic synapses at CCK+ connections in the hippocampus (*Daw et al., 2009*; *Lee et al., 2010*).

Another key finding was that endocannabinoid modulation of CCK+ connections depends on the postsynaptic target. Due to their distinct postsynaptic receptor expression profiles, IT and PT cells are known to respond differently to many neuromodulators, including dopamine, acetylcholine, and serotonin (*Anastasiades et al., 2019*; *Baker et al., 2018*; *Dembrow and Johnston, 2014*; *Shepherd, 2013*; *Stephens et al., 2018*). We observed robust endocannabinoid-mediated DSI at IT cells but not at nearby PT cells, despite the latter receiving stronger connections. Our result suggests the endocannabinoid system also shows marked cell-type specificity, with selective modulation on cortico-cortical networks. In the future, it will be important to assess the functional impact of this specificity on local processing in the PFC, including processing within and between hemispheres. It will also be important to establish if this specificity extends to other layers of the PFC, as IT cells are distributed from superficial L2 to deep L6 (*Anastasiades et al., 2019*).

In principle, selective DSI at CCK+ connections onto IT cells could reflect differences in the postsynaptic release or presynaptic detection of endocannabinoids. Our results are consistent with the latter explanation. First, our immunocytochemistry shows more CB1R puncta around IT cells, suggesting these receptors are selectively localized. This result was particularly surprising because we also found CCK+ interneurons make stronger connections onto PT cells. Second, direct activation of CB1Rs with WIN reduces CCK+-evoked IPSCs only at IT cells, with no effect at neighboring PT cells. This experiment bypasses the postsynaptic release of endocannabinoids, suggesting a presynaptic mechanism accounts for the specificity of DSI. The increase in PPR is also consistent with presynaptic modulation by CB1R, indicating reduced release probability (*Wilson et al., 2001*). By contrast, a recent study of differential DSI at CCK+ connections onto projection neurons in the amygdala suggested a post-synaptic mechanism, due to selective expression of endocannabinoid-synthesizing enzyme DGLα in pyramidal cells (*Vogel et al., 2016*). These findings indicate that different mechanisms leading to the modulation specificity may occur in different brain regions, underscoring the complexity of endocannabinoid systems.

The absence of WIN modulation at CCK+ connections onto PT cells, despite the low level of CB1R presence around their cell bodies, was surprising. Previous studies indicate that the nanoscale organization of CB1Rs relative to voltage-gated calcium channels influences the effectiveness of presynaptic CB1Rs (*Dudok et al., 2015*). This explains the weaker endocannabinoid modulation of dendritic-targeting than somatic-targeting CCK+ inputs in the hippocampus (*Lee et al., 2010*). Although we showed CCK+ interneurons restrict their targeting to the soma of both IT and PT cells, different nanoscale CB1R organizations could still occur at boutons, making the CB1R at CCK+ inputs onto PT neurons non-functional. Alternatively, some PT-targeting CB1R+ boutons may come from other interneuron subtypes, as previous studies showed that a small percentage of non-CCK+ cells also express CB1R (*Bodor et al., 2005*; *Marsicano and Lutz, 1999*).

A related question is whether CB1R+ and CB1R- connections arise from the same CCK+ interneurons. One possibility is that two different types of CCK+ interneurons exist: one that projects to

IT cells and is sensitive to CB1R modulation and another that projects to PT cells and is insensitive to CB1R modulation. Indeed, several subtypes of CCK+ interneurons are found across the brain, which differ in their subcellular targeting of postsynaptic cells, the size of their soma, and the expression of molecular markers such as VIP and VGLUT3 (*Bodor et al., 2005*; *del Pino et al., 2017*; *Lee et al., 2010*; *Omiya et al., 2015*; *Pelkey et al., 2020*; *Somogyi et al., 2004*). However, another possibility is that the same CCK+ interneuron projects to both projection neurons but the contacts at PT cells lack CB1R modulation (*Bodor et al., 2005*; *Dudok et al., 2015*; *Lee et al., 2010*). Distinguishing between these possibilities is challenging, but may be accomplished in the future using triple record- ings from CCK+ interneurons and pyramidal cells (*Reyes et al., 1998*), facilitated by soma-restricted optogenetic tools (*Collins et al., 2018*; *Mardinly et al., 2018*), which can be combined with our Dlx viral approach to specifically target CCK+ interneurons.

Ultimately, the ability of vHPC inputs to engage CCK+ interneurons, which in turn contact pyrami- dal cells, implicates a key role in feed-forward inhibition. Consistent with this idea, we observed prominent CB1R-mediated DSI of vHPC-evoked inhibition only at IT cells and not neighboring PT cells. Because there is no change in excitation, this modulation will selectively increase the excita- tion/inhibition (E/I) ratio at IT cells. In the intact brain, this could allow the vHPC to more effectively activate IT cells compared to neighboring PT cells. For example, when IT cells are highly active, increased endocannabinoid tone could promote local processing within the PFC. Disinhibition at IT cells could also allow for stronger responses to excitatory inputs, a potential mechanism for increased synchrony between hippocampus and PFC during working memory tasks and anxiety-like behavior (*Adhikari et al., 2010*; *Fujisawa and Buzsáki, 2011*; *O'Neill et al., 2013*). Furthermore, altered E/I balance could potentially shift the output of the PFC toward other intratelencephalic tar- gets throughout the brain, including other parts of the cortex, striatum, amygdala, and claustrum (*Anastasiades et al., 2019*; *Harris and Shepherd, 2015*).

Lastly, our results have implications for the functional properties of hippocampal-prefrontal net- works in health and disease (*Euston et al., 2012*; *Peters et al., 2010*; *Sierra-Mercado et al., 2011*; *Sotres-Bayon et al., 2012*). Previous studies have shown that endocannabinoid signaling can strongly influence both cognition and emotion (*Mechoulam and Parker, 2013*). Altering endocanna- binoid levels also affects executive function, working memory, stress, anxiety, and threat learning (*Lin et al., 2009*; *Lutz et al., 2015*; *Marcus et al., 2020*; *Marsicano et al., 2002*; *Volk and Lewis, 2016*). Our results indicate endocannabinoids influence communication between vHPC to PFC by selectively modulating connections from CCK+ interneurons to IT cells. In the future, it will be partic- ularly interesting to explicitly assess the role of this microcircuit and the impact of endocannabinoid modulation on PFC function and dysfunction, including threat learning and anxiety disorders (*Peters et al., 2010*; *Sierra-Mercado et al., 2011*; *Sotres-Bayon et al., 2012*).

# Materials and methods

**Key resources table**

| Reagent type (species) or resource | Designation | Source or reference | Identifiers | Additional information |
|---|---|---|---|---|
| Strain, strain background (*M. musculus*, both sexes) | C57BL/6J wild type | Jackson Labs | Stock #: 000664 RRID:IMSR_JAX:000664 | Both sexes |
| Strain, strain background (*M. musculus*, male) | *Pvalb*^tm1(cre)Arbr^ (PV-Cre) | Jackson Labs | Stock #: 008069 RRID:IMSR_JAX:008069 | Homozygote male breeder |
| Strain, strain background (*M. musculus*, male) | *Sst*^tm2.1(cre)Zjh^ (SOM-Cre) | Jackson Labs | Stock #: 013044 RRID:IMSR_JAX:013044 | Homozygote male breeder |
| Strain, strain background (*M. musculus*, male) | *Cck*^tm1.1(cre)Zjh^ (CCK-Cre) | Jackson Labs | Stock #: 012706 RRID:IMSR_JAX:012706 | Homozygote male breeder |
| Strain, strain background (*M. musculus*, female) | *Gt(ROSA)26 Sor*^tm14(CAG-tdTomato)Hze^ (Ai14) | Jackson Labs | Stock #: 007914 RRID:IMSR_JAX:007914 | Homozygote female breeder |

*Continued on next page*

*Continued*

| Reagent type (species) or resource | Designation | Source or reference | Identifiers | Additional information |
|---|---|---|---|---|
| Other | AAV1-DIO-ChR2-eYFP | UPenn | Cat #: AV-1–20298P | AAV virus expressing Cre-dependent ChR2 |
| Other | AAV1-ChR2-eYFP | UPenn | Cat #: AV-26973P | AAV virus expressing ChR2 |
| Other | AAV1-DIO-eYFP | UPenn | Cat #: AV-1–27056 | AAV virus expressing Cre-dependent eYFP |
| Other | AAVrg-GFP | Addgene | Cat #: 37825-AAVrg RRID:Addgene_37825 | Retrograde virus expressing GFP |
| Other | AAVrg-TdTomato | Addgene | Cat #: 59462-AAVrg RRID:Addgene_59462 | Retrograde virus expressing TdTomato |
| Other | AAV-Dlx-Flex-GFP | Addgene | Cat #: 83900 RRID:Addgene_83900 | AAV virus expressing Cre-dependent GFP in interneurons |
| Other | AAV-Dlx-Flex-ChR2-mCherry | This paper | | AAV virus expressing Cre-dependent ChR2 in interneurons |
| Antibody | Anti-PV (mouse, monoclonal) | Millipore | Cat#: MAB1572 RRID:AB_2174013 | (1:2000) |
| Antibody | Anti-CB1R (guinea pig, polyclonal) | Frontier Institute | Cat#: Af530 RRID:AB_2314113 | (1:500) |
| Antibody | Anti-SOM (rat, monoclonal) | Millipore | Cat#: MAB354 RRID:AB_2255365 | (1:200) |
| Chemical compound, drug | WIN 55,212–2 | Tocris | Cat#: 1038 | 1 µM |
| Chemical compound, drug | AM-251 | Tocris | Cat#: 1117 | 10 µM |

## Animals

Experiments used wild-type and transgenic mice of either sex in a C57BL/6J background (all breeders from Jackson Labs). Homozygote male breeders (PV-Cre = JAX 008069, SOM-Cre = JAX 013044, CCK-Cre = JAX 012706) were paired with female wild-type or Ai14 breeders (JAX 007914) to yield heterozygote offspring for experiments. All experimental procedures were approved by the University Animal Welfare Committee of New York University.

## Viruses

AAV viruses used in this study were as follows: AAV1.EF1a.DIO.hChR2(H134R)-eYFP.WPRE.hGH (UPenn AV-1–20298P), AAV1.hysn.hChR2(H134R)-eYFP.WPRE.hGH (UPenn AV-26973P), AAV1.EF1a. DIO.eYFP.WPRE.hGH (Upenn AV-1–27056), AAVrg.CAG.GFP (Addgene 37825-AAVrg), AAVrg. CAG.tdTomato (Addgene 59462-AAVrg). Additional viral constructs were assembled for Cre-dependent expression of a reporter under the control of the Dlx5/6 enhancer: AAV-Dlx-Flex-GFP (Addgene #83900) and AAV-Dlx-Flex-ChR2-mCherry (*Dimidschstein et al., 2016*). These constructs take advantage of the double-floxed inverted system, in which two consecutive and incompatible lox-sites are placed both in 5' and 3' of the reversed coding sequences of the viral reporter, restricting expression to Cre-expressing interneurons.

## Stereotaxic injections

Mice aged 4–6 weeks were deeply anesthetized with either isoflurane or a mixture of ketamine and xylazine, then head-fixed in a stereotaxic (Kopf Instruments). A small craniotomy was made over the injection site, using these coordinates relative to Bregma: PFC = ±0.4, –2.3, +2.1 mm, PAG = –0.6, both –2.5 and –3, –4.0 mm, vHPC = –3.3, both –3.6 and –4.2, –3 mm (mediolateral, dorsoventral, and rostrocaudal axes). For retrograde labeling, pipettes were filled with red retrogradely transported fluorescent beads (Lumafluor), Cholera Toxin Subunit B (CTB) conjugated to Alexa 647 (Life Technologies), or viruses. Borosilicate pipettes with 5 to 10 µm diameter tips were back-filled with

dye and/or virus, and a volume of 130–550 nl was pressure-injected using a Nanoject III (Drummond) every 30 s. The pipette was left in place for an additional 5 min, allowing time to diffuse away from the pipette tip, before being slowly retracted from the brain. For both retrograde and viral labeling, animals were housed for 2–3 weeks before slicing.

## Histology and fluorescence microscopy

Mice were anesthetized with a lethal dose of ketamine and xylazine, then perfused intracardially with 0.01 M phosphate-buffered saline (PBS) followed by 4% paraformaldehyde (PFA) in 0.01 M PBS. Brains were fixed in 4% PFA in 0.01 M PBS overnight at 4°C. Slices were prepared at a thickness of 70 µm for imaging intrinsic fluorescence or 40 µm for antibody staining (Leica VT 1000S vibratome). For immunohistochemistry, slices were incubated with blocking solution (1% bovine serum albumin and 0.2% Triton-X in 0.01 M PBS) for 1 hr at room temperature before primary antibodies were applied in blocking solution (mouse anti-parvalbumin antibody [Millipore, MAB1572] at 1:2000 overnight, rat anti-somatostatin [Millipore, MAB354] at 1:200 overnight, guinea pig anti-CB1R [Frontier Institute, Af530] at 1:500 for 36 hr) at 4°C. Slices were then incubated with secondary antibodies in blocking solution (goat anti-mouse 647 at 1:200, goat anti-rat 647 at 1:200, goat anti-guinea pig 647 at 1:500 [Invitrogen]) for 1.5 hr at room temperature before mounted under glass coverslips on gelatin-coated slides using ProLong Gold antifade reagent with DAPI (Invitrogen). Images were acquired using a confocal microscope (Leica SP8). Image processing involved adjusting brightness, contrast, and manual cell counting using ImageJ (NIH).

## Slice preparation

Mice aged 6–8 weeks were anesthetized with a lethal dose of ketamine and xylazine, and perfused intracardially with an ice-cold external solution containing the following (in mM): 65 sucrose, 76 NaCl, 25 NaHCO$_3$, 1.4 NaH$_2$PO$_4$, 25 glucose, 2.5 KCl, 7 MgCl$_2$, 0.4 Na-ascorbate, and 2 Na-pyruvate (295–305 mOsm), and bubbled with 95% O$_2$/5% CO$_2$. Coronal slices (300 µm thick) were cut on a VS1200 vibratome (Leica) in ice-cold external solution, before being transferred to ACSF containing (in mM): 120 NaCl, 25 NaHCO$_3$, 1.4 NaH$_2$PO$_4$, 21 glucose, 2.5 KCl, 2 CaCl$_2$, 1 MgCl$_2$, 0.4 Na-ascorbate, and 2 Na-pyruvate (295–305 mOsm), bubbled with 95% O$_2$/5% CO$_2$. Slices were kept for 30 min at 35°C, before being allowed to recover for 30 min at room temperature before starting recordings. All recordings were conducted at 30–32°C.

## Electrophysiology

Whole-cell recordings were obtained from pyramidal neurons or interneurons located in layer 5 (L5) of infralimbic (IL) PFC. Neurons were identified by infrared-differential interference contrast or fluorescence, as previously described (*Chalifoux and Carter, 2010*). In the case of pyramidal cells, the projection target was established by the presence of retrobeads or Alexa-conjugated CTB, as previously described (*Little and Carter, 2013*). Pairs of adjacent cells were chosen for sequential recording, ensuring they received similar inputs (typically < 50 µm between cells). Borosilicate pipettes (2–5 MΩ) were filled with internal solutions. Three types of recording internal solutions were used. For current-clamp recordings (in mM): 135 K-gluconate, 7 KCl, 10 HEPES, 10 Na-phosphocreatine, 4 Mg$_2$-ATP, and 0.4 Na-GTP, 290–295 mOsm, pH 7.3, with KOH. For voltage-clamp recordings (in mM): 135 Cs-gluconate, 10 HEPES, 10 Na-phosphocreatine, 4 Mg$_2$-ATP, and 0.4 Na-GTP, 0.5 EGTA, 10 TEA-chloride, and 2 QX314, 290–295 mOsm, pH 7.3, with CsOH. For DSI experiments (in mM): 130 K-gluconate, 1.5 MgCl$_2$, 10 HEPES, 1.1 EGTA, 10 phosphocreatine, 2 MgATP, 0.4 NaGTP. In some experiments studying cellular morphology, 5% biocytin was also included in the recording internal solution. After allowing biocytin to diffuse through the recorded cell for at least 30 min, slices were fixed with 4% PFA before staining with streptavidin conjugated to Alexa 647 (Invitrogen).

Electrophysiology recordings were made with a Multiclamp 700B amplifier (Axon Instruments), filtered at 4 kHz for current-clamp, and 2 kHz for voltage-clamp, and sampled at 10 kHz. The initial series resistance was <20 MΩ, and recordings were ended if series resistance rose above 25 MΩ. In some experiments, 1 µM TTX was added to block action potentials, and 100 µM 4-AP and 4 mM external Ca$^{2+}$ to restore presynaptic release. In many experiments, 10 µM CPP was used to block NMDA receptors. In current-clamp experiments characterizing intrinsic properties, 10 µM NBQX, 10 µM CPP, and 10 µM gabazine were used to block excitation and inhibition. In some experiments, 10

μm AM-251 was used to block CB1 receptors or 1 μM WIN 55,212–2 was used to activate CB1 receptors. All chemicals were purchased from either Sigma or Tocris Bioscience.

## Optogenetics

Channelrhodopsin-2 (ChR2) was expressed in presynaptic neurons and activated with a brief light pulse from a blue LED (473 nm) (Thorlabs). For wide-field illumination, light was delivered via a 10 × 0.3 NA objective (Olympus) centered on the recorded cell. LED power was routinely calibrated at the back aperture of the objective. LED power and duration were adjusted to obtain reliable responses, with typical values of 0.4 to 10 mW and 2 ms, respectively. Subcellular targeting experiments were performed with a Polygon DMD device (Mightex) focused through a 10 × 0.3 NA objective (Olympus) with a 75 μm pixel size. Pulses were delivered at 1 Hz using a pseudo-random 10 × 10 grid pattern, yielding an effective mapping area of 750 μm × 750 μm. Experiments used a 2 ms LED pulse yielding an effective power of 0.17 mW per pixel.

## Data analysis

Electrophysiology and imaging data were acquired using National Instruments boards and MATLAB (MathWorks) (*Pologruto et al., 2003*). Off-line analysis was performed using Igor Pro (WaveMetrics). Intrinsic properties were determined as follows. Input resistance was calculated from the steady-state voltage during a −50 pA, 500 ms current step. Voltage sag ratio was calculated as $(V_{sag} - V_{ss})$ / $(V_{sag} - V_{baseline})$, where $V_{sag}$ is average over a 1 ms window around the minimum, $V_{ss}$ is average of last 50 ms, and $V_{baseline}$ is average of 50 ms preceding the current injection. The membrane time constant (tau) was measured using exponential fits to these hyperpolarizations. Adaptation was calculated as the ratio of the first and last inter-spike intervals, such that a value of 1 indicates no adaptation and values < 1 indicate lengthening of the inter-spike interval. For experiments with a single optogenetic stimulation, the PSC amplitude was measured as the average value across 1 ms around the peak subtracted by the average 100 ms baseline value prior to the stimulation. For experiments with a train of optogenetic stimulation, each PSC amplitude was measured as the average value in a 1 ms window around the peak, minus the average 2 ms baseline value before each stimulation. Most summary data are reported in the text and figures as arithmetic mean ± SEM. Ratios of responses at pairs of cells are reported as geometric mean in the text, and with ± 95% confidence interval (CI) in the figures, unless otherwise noted. Comparisons between unpaired data were performed using non-parametric Mann-Whitney test. Comparisons between data recorded in pairs were performed using non-parametric Wilcoxon test. Two-tailed p values < 0.05 were considered significant.

## Acknowledgements

We thank the Carter lab for helpful discussions and comments on the manuscript. This work was supported by NIH R01 MH085974 (AGC). The authors have no financial conflicts of interest.

## Additional information

### Funding

| Funder | Grant reference number | Author |
| --- | --- | --- |
| National Institute of Mental Health | R01 MH085974 | Adam G Carter |

The funders had no role in study design, data collection and interpretation, or the decision to submit the work for publication.

### Author contributions

Xingchen Liu, Conceptualization, Data curation, Formal analysis, Investigation, Writing - original draft, Writing - review and editing; Jordane Dimidschstein, Gordon Fishell, Resources, Methodology, Writing - review and editing; Adam G Carter, Conceptualization, Supervision, Funding acquisition, Writing - original draft, Project administration, Writing - review and editing

**Author ORCIDs**

Xingchen Liu (ID) https://orcid.org/0000-0002-7867-3243

Gordon Fishell (ID) http://orcid.org/0000-0002-9640-9278

Adam G Carter (ID) https://orcid.org/0000-0003-2095-3901

**Ethics**

Animal experimentation: This study was performed in strict accordance with the recommendations in the Guide for the Care and Use of Laboratory Animals of the National Institutes of Health. All of the animals were handled according to approved university animal welfare committee (UAWC) protocols (#07-1281) of New York University.

**Decision letter and Author response**

Decision letter https://doi.org/10.7554/eLife.55267.sa1

Author response https://doi.org/10.7554/eLife.55267.sa2

## Additional files

**Supplementary files**

• Transparent reporting form

**Data availability**

All data generated or analyzed during this study are included in the manuscript and supporting files.

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
