## [Decision Letter]

**Acceptance summary:**

This is a carefully designed and executed study that illuminates some of the inhibitory circuitry, short-term plasticity and modulation present at the connections linking ventral hippocampus and medial prefrontal cortex in the mouse. These connections are known to be important for working memory and other behaviors and so details of the underlying circuitry are likely to be important for our understanding of the contributions of these brain regions to complex behaviors in mammals. In addition, the finding that cannabinoid-mediated DSI is specific to one set of inhibitory outputs adds a new level of refinement to how we think about this well studied form of synaptic plasticity/modulation, and hence the role of this prominent GPCR in cortical physiology.

**Decision letter after peer review:**

Thank you for submitting your article "Hippocampal inputs engage CCK+ interneurons to mediate endocannabinoid-modulated feed-forward inhibition in the prefrontal cortex" for consideration by *eLife*. Your article has been reviewed by three peer reviewers, including Sacha B Nelson as the Reviewing Editor and Reviewer #1, and the evaluation has been overseen by Gary Westbrook as the Senior Editor. The following individuals involved in review of your submission have agreed to reveal their identity: Chris J McBain (Reviewer #2); Pablo Castillo (Reviewer #3). The reviewers have discussed the reviews with one another and the Reviewing Editor has drafted this decision to help you prepare a revised submission.

Summary:

This is an elegant study that illuminates some of the inhibitory circuitry, short-term plasticity and modulation present at the connections linking ventral hippocampus and medial prefrontal cortex in the mouse. These connections are known to be important for working memory and other behaviors and so details of the underlying circuitry are likely to be important for our understanding of the contributions of these brain regions to complex behaviors in mammals. The data are of high quality and the results are compelling and present an important advance in our understanding of CCK driven inhibition in the PFC. In addition, the finding that cannabinoid-mediated DSI is specific to one set of inhibitory outputs adds a new level of refinement to how we think about this well studied form of synaptic plasticity/modulation, and hence the role of this prominent GPCR in cortical physiology.

Essential revisions:

1) The stated author contributions do not meet the ICMJE recommendations for authorship. See: http://www.icmje.org/recommendations/browse/roles-and-responsibilities/defining-the-role-of-authors-and-contributors.html and https://submit.elifesciences.org/html/elife_author_instructions.html#

2) Figure 5 and related text. The observation that DSI is prominent at IT connections but not PT connections is interesting and warrants further consideration. The authors suggest that a lower density of CB1R-positive boutons at inputs onto PT (5.7 puncta) versus IT (9.3 puncta) may explain this discrepancy. However this observation seems overly simplistic given the previous STORM study of Dudok et al., 2014, that describe a relatively uniform density of CB1Rs/bouton at somatic versus dendritic targets despite Lee et al., 2010, describing prominent DSI at somatic inputs versus dendritic inputs suggesting a more complex interplay between CB1Rs and downstream machinery that can be anticipated from CB1R distribution alone. Can the authors speculate further about their lack of DSI at PT connections? Do all CCK inputs tested innervate only the somatic compartment of their pyramidal cell counterparts? If there are 5.7 CB1R-positive puncta at CCK-interneuron inputs to PT cells why are they resistant to exogenous application of the CB1R agonist WIN?

Related to this, some figures (schemes Figures 4D and 7F) suggest that the same CCK+ interneuron projects to PT and IT cells. Do the authors know whether this is the case? Two types of CCK+ interneurons may exist, those expressing CB1Rs and projecting to IT cells, and those lacking CB1Rs and targeting PT cells. The authors seem to assume that CB1R labeling in Figure 6A, B arises from CCK+ interneurons but there is little experimental evidence in support of this possibility. Actually, CB1R labeling in the soma of PT and IT cells may arise from non-CCK+ interneurons. If CCK+ Int-PT connections do express CB1Rs, then, the experiments reported in Figure 6D, E may indicate that these receptors are non-functional. The authors may want to experimentally resolve these issues, or properly discuss these potential scenarios.

3) Figure 4 and related text. Is the apparent smaller amplitude of the inhibitory input to IT PNs compared to PT PNs a result of tonic cannabinoid inhibition? In the experiments shown in Figure 5 the application of the AM-251 inverse agonist appears to boost the IPSC amplitude during the wash epoch in the example shown (but not in the data shown in Figure 6). Do the authors see a consistent boost of the IPSC at IT cells on application of AM-251 alone (it is unclear from the data shown in Figure 6 whether AM-251 was added on its own or only after WIN application)?

4) Figure 7 and related text. Although the authors place their emphasis on the vHPC driven CCK inputs onto PT versus IT cells can they make an estimation of the relative drive by PV:CCK by dissecting these inputs using agatoxin versus conotoxin to selectively block PV versus CCK inputs?

5) DSI is a rather non-physiological way of mobilizing endocannabinoids. It would be important to know whether a more physiological manipulation, i.e. IT cell firing, also suppresses GABAergic inputs from CCK+ interneurons. Also, DSI is presented in a rather unconventional manner by reporting three time points only (before, after recovery), whereas most studies show the time course of this phenomenon by testing synaptic transmission every 3-5 seconds. Most studies report changes in PPR associated with the synaptic suppression. The representative traces reported in Figure 5C suggest little to no change in PPR. Is this the case? Provide a PPR summary plot (before, after, recovery).

---

## [Author Response]

Essential revisions:1) The stated author contributions do not meet the ICMJE recommendations for authorship. See: http://www.icmje.org/recommendations/browse/roles-and-responsibilities/defining-the-role-of-authors-and-contributors.html and https://submit.elifesciences.org/html/elife_author_instructions.html#

We have specified the roles of each author in the author contributions and confirmed that all authors meet the ICMJE authorship recommendations. Please note that Jordane Dimidschstein and Gord Fishell developed unpublished reagents, including AAV-Dlx-Flex-ChR2 to target CCK+ interneurons in CCK-Cre mice, without which this paper would not have been possible.

2) Figure 5 and related text. The observation that DSI is prominent at IT connections but not PT connections is interesting and warrants further consideration. The authors suggest that a lower density of CB1R-positive boutons at inputs onto PT (5.7 puncta) versus IT (9.3 puncta) may explain this discrepancy. However this observation seems overly simplistic given the previous STORM study of Dudok et al., 2014, that describe a relatively uniform density of CB1Rs/bouton at somatic versus dendritic targets despite Lee et al., 2010, describing prominent DSI at somatic inputs versus dendritic inputs suggesting a more complex interplay between CB1Rs and downstream machinery that can be anticipated from CB1R distribution alone.Can the authors speculate further about their lack of DSI at PT connections?

This is an important point, which we now address in the text and Author response image 1. Please note we first observed stronger CCK+ inputs onto PT cells (Figure 4C-D), but more CB1R labeling at the cell bodies of IT cells (Figure 4H-J). The latter result suggested there might be differences in DSI at these projection neurons, and we observed more DSI at IT cells (Figure 5). The next question was whether this reflected differences in EC release (postsynaptic) or CB1R sensitivity (presynaptic). Supporting the latter possibility, we found that wash-in of AM-251 reduces CCK+-evoked IPSCs only at IT cells (Figure 6). The way the text was originally written made it seem like the CB1R labeling was an explanation, when really it was a hint that modulation was different. We have therefore rearranged both the text and the figures to make our data and interpretation clearer.

**Author response image 1. sa2fig1:** Quantification of CB1R+ and CCK+ puncta around IT and PT soma. A) Injection schematic. AAVrg-TdTomato was injected to either the cPFC or the Pag along with AAV-Dlx-Flex-GFP to the PFC in CCK-Cre animals, to label IL IT or PT cells with Td-Tomato and CCK+ axons with GFP. Slices were subsequently stained with anti-CB1R antibody. B) Left, Images of GFP and CB1R staining around PT cells. Scale bar = 10 µm. Right, Quantification of average CB1R+, GFP+ and co-labeled puncta (duo+) numbers around IT cells. Each gray line links the counts from one cell. (n = 82 cells, 3 animals) C) Similar to (B), for IT cells. (n = 71 cells, 3 animals) D) Left, Quantification schematic. Right, The percentages of co-labeled puncta (duo+) among GFP+ puncta for IT and PT cells.

We agree that lower CB1R density around IT and PT cells does not fully explain the difference in DSI and AM-251 results. After all, there are CB1R around the soma of PT cells, yet we consistently do not see modulation in our experiments (Figures 5, 6 and 7). We think it is unlikely that differential modulation reflects somatic versus. dendritic targeting, as we have conducted new experiments that indicate that CCK+ inputs are predominantly somatic onto both IT and PT cells (Figure 4—figure supplement 1; more details below). However, we agree it may reflect the nanoscale organization of coupling between presynaptic CB1Rs and the release machinery, and we have revised our Discussion to address these points and discuss other potential mechanisms (Discussion, eighth paragraph).

Do all CCK inputs tested innervate only the somatic compartment of their pyramidal cell counterparts?

We have performed new sCRACM experiments to study the sub-cellular organizations of CCK+ inputs to pyramidal cells (Figure 4—figure supplement 1). As in our other experiments, we first expressed ChR2 in CCK+ cells and retrogradely labeled IT and PT cells. We then isolated monosynaptic connections by including 1 µM TTX to block APs and 10 µM 4-AP to recover release. We mapped a 75 µm spot of light across the entire dendritic arbor and recorded evoked IPSCs at the cell body. Note that we have previously used this approach to map IPSCs at the soma or dendrites of pyramidal cells (Marlin and Carter, 2014). In this case, we found that CCK+ inputs are highly restricted to the soma of both IT cells (Figure 4—figure supplement 1A-B) and PT cells (Figure 4—figure supplement 1C-D). While there are some inputs to the dendrites of both cell types, there are no major differences in dendritic targeting. We have described this new experiment in the revised text (subsection “CCK+ interneurons make connections onto L5 pyramidal cells).

If there are 5.7 CB1R-positive puncta at CCK-interneuron inputs to PT cells why are they resistant to exogenous application of the CB1R agonist WIN?

We speculate that presynaptic CB1Rs at CCK+ inputs onto PT cells may be non-functional, perhaps due to uncoupling of these presynaptic receptors and the release machinery, as described by (Dudok et al., 2015). We have included this point in the Discussion (eighth paragraph).

Related to this, some figures (schemes Figures 4D and 7F) suggest that the same CCK+ interneuron projects to PT and IT cells. Do the authors know whether this is the case? Two types of CCK+ interneurons may exist, those expressing CB1Rs and projecting to IT cells, and those lacking CB1Rs and targeting PT cells.

This is an important point, and we have now changed the schematics in Figures 4 and 7 to reflect the fact that we cannot be sure that the same CCK+ interneuron contacts IT and PT cells.

We agree it is possible that at least two types of CCK+ interneurons exist and may differentially contact IT and PT cells. In other parts of cortex and hippocampus, subtypes of CCK+ interneurons are classified by intrinsic properties, morphologies, and molecular markers such as VGLUT3 and VIP, and soma sizes (Bodor et al., 2005; Daw et al., 2009; Del Pino et al., 2017; Lee et al., 2010; Omiya et al., 2015; Pelkey et al., 2020; Somogyi et al., 2004). It is certainly possible that one subtype of CCK+ interneurons primarily contacts IT cells and undergoes CB1R modulation, and another subtype contacts PT cells and does not undergo CB1R modulation. Alternatively, individual cells in the same class could make biased connections onto IT and PT cells, or selectively express CB1Rs at the presynaptic terminals of these connections.

Unfortunately, answering this question proved to be very difficult. We tried to use intersectional genetic tools to separate CCK+ interneurons, but found that the viral tools did not work, and the triple transgenic mice would take too long to establish. We then tried to use triple recordings from CCK+ interneurons, IT cells and PT cells, but the yield was too low, and it was impossible to do these experiments in a timely fashion. We also tried optogenetics, expressing a soma-tagged opsin (st-Chrome) in CCK+ cells, but because we do not have a Dlx version, there was contamination from pyramidal cells. We think this is an extremely interesting question, but will require new technologies to answer, and we think it is beyond the scope of this paper. However, we have now provided additional discussion on this point in the revised manuscript (Discussion, ninth paragraph).

The authors seem to assume that CB1R labeling in Figure 6A, B arises from CCK+ interneurons but there is little experimental evidence in support of this possibility. Actually, CB1R labeling in the soma of PT and IT cells may arise from non-CCK+ interneurons. If CCK+ Int-PT connections do express CB1Rs, then, the experiments reported in Figure 6D, E may indicate that these receptors are non-functional. The authors may want to experimentally resolve these issues, or properly discuss these potential scenarios.

We agree that CB1R+ labeling around the soma of IT and PT cells could also arise from non-CCK+ interneurons. To address this concern, we performed further immunohistochemical staining experiments, as shown in Author response image 1. CCK+ axons were labeled with GFP, PT or IT cells were retrogradely labeled with tdTomato, and CB1Rs were stained. GFP+ and CB1R+ puncta and their overlap (Duo+) were quantified around the soma of PT and IT cells (Author response image 1 – C). As before, we found more CB1R+ puncta at IT cells (Author response image 1 and C), indicating this is a robust result (see also Figure 4). However, we did not observe differences in either GFP+ puncta or Duo+ at IT and PT cells (Author response image 1 and C). One caveat is the variable expression of AAV-Dlx-Flex-GFP virus, which will differ between animals. To address this concern, we also normalized the number of CB1R+GFP+ (Duo+) puncta to the number of GFP+ puncta and found co-labeling of 45% at PT cells and 55% at IT cells, which was a small but significant difference (Author response image 1). Because we saw no CB1R modulation of CCK+ inputs onto PT cells, this suggests these CB1R+ CCK+ innervations are non-functional. On the other hand, the presence of GFP- CB1R+ puncta could be due to either low efficiency of the viral labeling, or the possibility that they belong to other interneurons. We found these results inconclusive and have not included them in the paper. Instead, we have addressed these points in our revised Discussion, as suggested (eighth paragraph).

3) Figure 4 and related text. Is the apparent smaller amplitude of the inhibitory input to IT PNs compared to PT PNs a result of tonic cannabinoid inhibition? In the experiments shown in Figure 5 the application of the AM-251 inverse agonist appears to boost the IPSC amplitude during the wash epoch in the example shown (but not in the data shown in Figure 6). Do the authors see a consistent boost of the IPSC at IT cells on application of AM-251 alone (it is unclear from the data shown in Figure 6 whether AM-251 was added on its own or only after WIN application)?

We performed new experiments to test whether CCK+ inputs onto IT cells are subject to tonic CB1R modulation (Figure 6—figure supplement 1). We recorded CCK+-evoked IPSCs in IT cells at -50mV, obtaining a stable baseline, then washing in 10 µM AM-251 (Figure 6—figure supplement 1A and B). We observed a slight but significant increase of IPSCs after 10 minutes (Figure 6—figure supplement 1B and C; IPSC amplitude ratio AM-251 / baseline = 1.18 ± 0.10, p = 0.01). This indicates weak tonic CB1R modulation on IT cells. We have included this experiment in the revised text (subsection “Endocannabinoid modulation depends on postsynaptic cell type”). However, this tonic CB1R modulation does not explain the large difference between baseline CCK+ inputs onto IT and PT cells (Figure 4D, IPSC amplitude ratio PT/IT = 4.3 ± 0.9). Instead, we think this is likely due to biased inhibition, which we also observe for PV+, SOM+ and NDNF+ inputs onto IT and PT cells (Anastasiades et al., 2020; Anastasiades et al., 2018).

We are not sure about the comment on Figure 5, as the data in 5C and 5D were acquired in different cells, and there is no wash epoch. The latter is a separate set of experiments to test if any DSI remained in AM-251, so the amplitudes in 5C and 5D cannot be directly compared.

4) Figure 7 and related text. Although the authors place their emphasis on the vHPC driven CCK inputs onto PT versus IT cells can they make an estimation of the relative drive by PV:CCK by dissecting these inputs using agatoxin versus conotoxin to selectively block PV versus CCK inputs?

The relative contribution of PV+ and CCK+ interneurons in vHPC-mediated feedforward inhibition is an interesting point. Our data in Figure 2 suggest vHPC activates both populations, but unfortunately, we do not have the quantitative measurement of the relative contribution.

Unfortunately, using agatoxin and conotoxin will not only block the release of PV+ and CCK+ interneurons, they also affect the vHPC inputs onto these interneurons, making this method unspecific in dissecting PV+ and CCK+ inputs in the context of feedforward inhibition.

Another way to determine the relative contribution of PV+ and CCK+ interneurons is to directly record the vHPC inputs at these interneurons in the same animal and slice. To achieve this, one could cross CCK-Cre and PV-Flp mice, use CreOn or FlpOn fluorophores to label each population, and perform pair-recordings to determine the relative inputs of vHPC onto PV+ and CCK+ interneurons. However, due to the restrictions of animal ordering and breeding during the COVID-19 pandemic, as well as the need to validate the new PV-Flp mice and new FlpOn viruses, we have not yet been able to perform this experiment. If the reviewers feel it is essential, we have just ordered the mice, but it may take another half year to get the results.

Overall, we think the answer of the relative drive onto PV+ or CCK+ would be certainly interesting but is not crucial to the main findings of this paper. Thus, we decided to submit the revised manuscript without this experiment and include this point in the Discussion (fourth paragraph).

5) DSI is a rather non-physiological way of mobilizing endocannabinoids. It would be important to know whether a more physiological manipulation, i.e. IT cell firing, also suppresses GABAergic inputs from CCK+ interneurons.

This is an interesting point, and we performed new experiments to test IT cell firing-induced suppression of CCK+ inputs (Figure 5—figure supplement 3). This experiment was similar to our voltage-clamp DSI experiment shown in Figure 5, except that cells were now recorded in current-clamp, and the “depolarization” was 5 s of 20 Hz APs induced by 5 ms 2 nA step current pulses (Figure 5—figure supplement 3A). We observed a pronounced reduction of CCK+ IPSPs at IT neurons after the AP train (Figure 5—figure supplement 3B), suggesting possible DSI. However, some of the reduction persisted in the presence of AM-251 (Figure 5—figure supplement 3C), suggesting other factors may contribute. In both sets of experiments, we also noticed an inevitable hyperpolarization of the resting membrane potential after the AP train, which could potentially reduce the IPSP driving force and confound the conclusion of IPSP reduction by CB1R mobilization. To account for this possibility, we also recorded CCK+ IPSPs at the matched hyperpolarized potential for each cell (Vm corrected baseline input) (Figure 5—figure supplement 3B and C). In control conditions, we found that hyperpolarization reduced the IPSP amplitude, but not as much as the AP train (Figure 5—figure supplement 3D), confirming that some of the reduction of CCK+ IPSPs may indeed reflect genuine DSI. Consistent with this interpretation, there was also no difference between the effects of hyperpolarization and the AP train in the presence of AM-251 (Figure 5—figure supplement 3E). We have included this new experiment in the revised manuscript (subsection “CB1R-mediated DSI depends on the postsynaptic cell-type”, last paragraph).

Also, DSI is presented in a rather unconventional manner by reporting three time points only (before, after recovery), whereas most studies show the time course of this phenomenon by testing synaptic transmission every 3-5 seconds. Most studies report changes in PPR associated with the synaptic suppression. The representative traces reported in Figure 5C suggest little to no change in PPR. Is this the case? Provide a PPR summary plot (before, after, recovery).

We have performed new experiments to address the time course of DSI recovery (Figure 5—figure supplement 1). Optogenetic stimulation of CCK+ inputs was performed every 5 seconds during the 30 s baseline period and the 60 s recovery period after the depolarization (Figure 5—figure supplement 1A). We observed a gradual recovery of IPSC amplitudes after the depolarization (Figure 5—figure supplement 1B and C), consistent with many previous studies (Glickfeld and Scanziani, 2006; Wilson and Nicoll, 2001). We have included this experiment in the revised manuscript (subsection “CB1R-mediated DSI depends on the postsynaptic cell-type”).

The reviewer raises a good point about PPR, and we show a summary plot in Author response image 2. Contrary to our expectations, we did not see a significant PPR change after depolarization, which could be due to using optogenetics rather than electrical stimulation.

**Author response image 2. sa2fig2:** Quantification of PPR. Quantification of PPR of CCK+ inputs at IT cells before (black), after (red) and recovery (gray) from depolarization in the DSI experiment (n = 9 cells, 5 animals).